# Redox-powered autonomous directional C–C bond rotation under enzyme control

Jordan Berreur[1], Olivia F. B. Watts[1,2], Theo H. N. Bulless[1,2], Nicholas T. O'Donoghue[1,2], Marc Del Olmo[1,2], Ashley J. Winter[1], Jonathan Clayden[1✉] & Beatrice S. L. Collins[1✉]

Living biological systems rely on the continuous operation of chemical reaction networks. These networks sustain out-of-equilibrium regimes in which chemical energy is continually converted into controlled mechanical work and motion[1–3]. Out-of-equilibrium reaction networks have also enabled the design and successful development of artificial autonomously operating molecular machines[4,5], in which networks comprising pairs of formally—but non-microscopically—reverse reaction pathways drive controlled motion at the molecular level. In biological systems, the concurrent operation of several reaction pathways is enabled by the chemoselectivity of enzymes and their cofactors, and nature's dissipative reaction networks involve several classes of reactions. By contrast, the reactivity that has been harnessed to develop chemical reaction networks in pursuit of artificial molecular machines is limited to a single reaction type. Only a small number of synthetic systems exhibit chemically fuelled continuous controlled molecular-level motion[6–8] and all exploit the same class of acylation–hydrolysis reaction. Here we show that a redox reaction network, comprising concurrent oxidation and reduction pathways, can drive chemically fuelled continuous autonomous unidirectional motion about a C–C bond in a structurally simple synthetic molecular motor based on an achiral biphenyl. The combined use of an oxidant and reductant as fuels and the directionality of the motor are both enabled by exploiting the enantioselectivity and functional separation of reactivity inherent to enzyme catalysis.

Continuous directional rotation about an axis underpins macroscopic machinery and will probably prove crucial for future nanoscale machines. Synthetic molecular-scale rotary motors featuring light-driven rotation about a C=C double bond have been pioneered by the Feringa group over the past 25 years (refs. 9,10). Autonomous chemically fuelled directional rotation about a C–N single bond has been reported[8,11–14], driven by opposing acylation and ester hydrolysis pathways that form a cyclic reaction network that is coupled to the exergonic hydrolysis of a carbodiimide fuel. Similar acylation–hydrolysis reaction networks have been used to achieve directional translational motion and to fuel out-of-equilibrium supramolecular assemblies[5,6,15–17], but the lack of synthetic chemically fuelled molecular motors driven by alternative reaction networks reflects the challenge of finding and using suitable mutually compatible opposing reactions.

All life depends on redox chemistry, but despite the evolution in nature of several systems that allow reduction and oxidation pathways to run concurrently, cyclic redox reaction networks have not been used to drive unidirectional motion in artificial molecular motors. In this paper, we present a molecular motor in which autonomous unidirectional rotation about a C–C single bond is driven by a cyclic redox reaction network. Concurrent biocatalytic oxidation and chemical reduction reactions of a biphenyl motor create a cyclic reaction network that consumes simple fuels (oxygen and borane) to drive rotary motion

about a C–C bond, with directionality governed by the enantioselectivity of the biocatalytic oxidation.

Our design builds on an archetypal cyclic reaction network in synthetic chemistry: cyclic deracemization[18–22]. Cyclic deracemizations enable the contra-thermodynamic enrichment of one enantiomer from a racemic mixture through the operation of a dissipative cyclic reaction network. For example, in the seminal redox-driven cyclic deracemization from Turner and co-workers shown in Fig. 1a (ref. 23), enantioselective oxidation of racemic α-methylbenzylamine is coupled with non-stereoselective reduction, leading to enrichment in the less reactive of the two amine enantiomers, (R)-α-methylbenzylamine. Concurrent oxidation and reduction is achieved by recourse to an oxidation biocatalyst (monoamine oxidase) that can function in the presence of a reducing agent, in this case, ammonia borane ($H_3N·BH_3$).

A related, but as yet underexplored, model for cyclic deracemization passes not through a transient achiral intermediate (the imine in Fig. 1a) but instead through a transient state that consists of a pair of rapidly interconverting enantiomers[24]. A model for such a deracemization is shown in Fig. 1b. Oxidation of the chiral atropisomeric diol **1** would give hydroxyaldehyde **2**, which is expected to racemize rapidly at room temperature owing to a bonding interaction between the phenolic hydroxyl group and the aldehyde carbonyl (shown in square brackets). Covalent bonding interactions that lower energy barriers to

[1]School of Chemistry, University of Bristol, Bristol, UK. [2]These authors contributed equally: Olivia F. B. Watts, Theo H. N. Bulless, Nicholas T. O'Donoghue, Marc Del Olmo. ✉e-mail: j.clayden@bristol.ac.uk; bs.lefanucollins@bristol.ac.uk

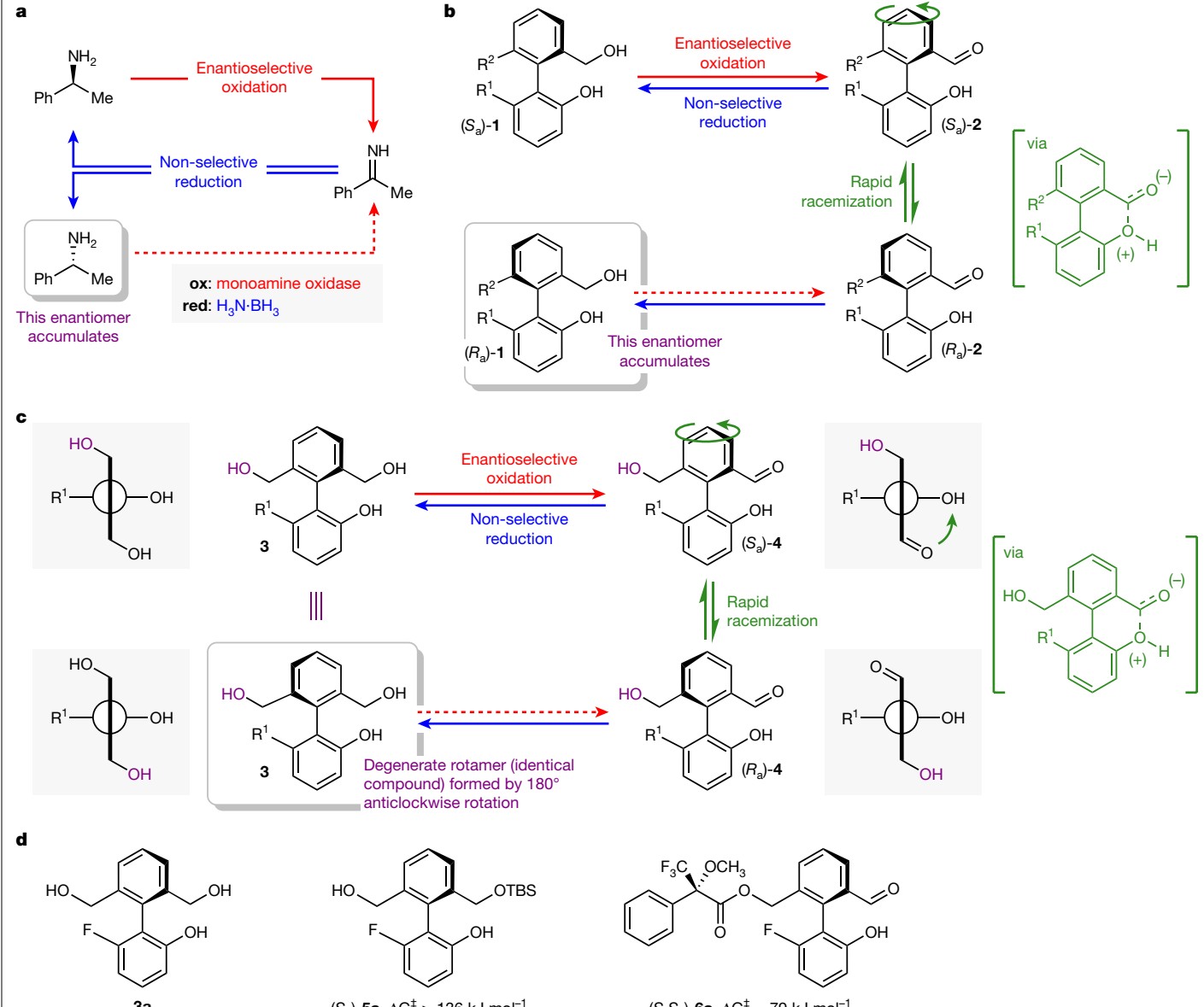

**Fig. 1 | Cyclic deracemization as a model for unidirectional rotation. a**, Cyclic deracemization of a chiral amine by way of an achiral imine, coupling enantioselective biocatalytic oxidation with non-selective reduction using ammonia borane[23]. **b**, Cyclic deracemization of an atropisomeric biphenyl by way of a pair of rapidly interconverting enantiomeric conformers;

$R^1, R^2$ = unspecified substituents. **c**, Design for a unidirectional rotary molecular motor; $R^1$ = unspecified substituent. **d**, Motor **3a** and derivatives of **3a**, $(S_a)$-**5a** and $(S,S_a)$-**6a**. Stereochemical assignment assumes that $R^1$ has higher priority than OH and $R^2$ has lower priority than $CH_2OH$.

bond rotation provide the basis for Bringmann's 'lactone' method and other more recent strategies for the enantioselective synthesis of atropisomers[25,26], and related covalent bonding interactions are exploited in the rotary motor of Leigh and co-workers and other stepwise rotary molecular motors[8,27–30]. Transient non-covalent bonding interactions[31], as proposed for hydroxyaldehyde **2**, have been used extensively in the asymmetric synthesis of atropisomers through dynamic kinetic resolution methods using organocatalysis, transition-metal catalysis and enzymatic catalysis[26,32,33].

In analogy to Turner's cyclic deracemization of a point chiral centre[23] (Fig. 1a), deracemization of atropisomeric **1** occurs if the oxidation to hydroxyaldehyde **2** proceeds enantioselectively and is coupled to a non-selective reduction back to the diol **1**. Such an atropisomeric deracemization is also accompanied by net directional motion: if the oxidation of **1** to **2** is enantioselective, the rapidly interconverting mixture of enantiomeric conformers of **2** is approached selectively from

one direction, and every transformation of $(S_a)$-**1** to $(R_a)$-**1** results in a 180° anticlockwise rotation of the upper ring, as viewed from above, indicated by the curved arrow.

In Fig. 1c, we outline how one small change to the biphenyl deracemization substrate **1** reveals a simple design for a unidirectional rotary molecular motor. The same cyclic reaction network is applied to closely analogous molecule **3**, which has two hydroxymethyl substituents in the *ortho* positions of the upper ring (Fig. 1c, in which $R^2 = CH_2OH$). Triol **3** is now achiral by virtue of the plane of symmetry that bisects the upper ring. Enantioselective oxidation desymmetrizes triol **3** to give an enantioenriched sample of chiral monoaldehyde $(S_a)$-**4** that itself undergoes rapid racemization owing to the bonding interaction between the phenolic hydroxyl group and the aldehyde carbonyl (shown in square brackets). From monoaldehyde **4**, the concurrent non-selective reduction does not return the enantiomer of the starting triol, because this achiral triol is, by definition, superimposable on its mirror image.

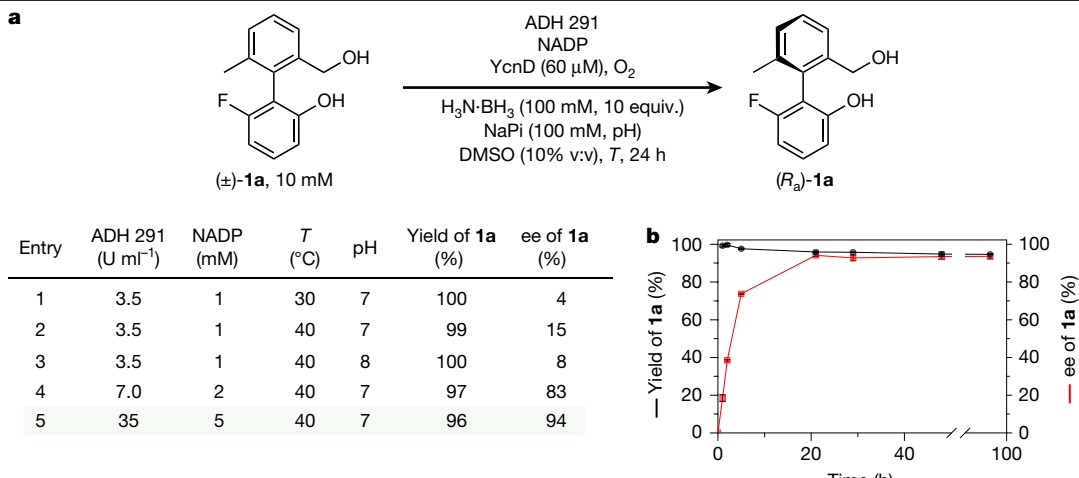

**Fig. 2 | Deracemization of biaryl 1a. a**, Optimization of deracemization of **1a**: (±)-**1a** (10 mM), ADH 291, NADP, NADPH oxidase (YcnD (60 μM)), NaPi (100 mM), DMSO (10% v:v), H$_3$N·BH$_3$ (100 mM, 10 equiv.), O$_2$ (air), ee analysis performed by high-performance liquid chromatography at 24 h. **b**, Yield and ee of **1a** over time for conditions in entry 5; error bars are derived from triplicate reactions, in which the error given is the sample standard deviation between the triplicates.

| Entry | ADH 291 (U ml$^{-1}$) | NADP (mM) | $T$ (°C) | pH | Yield of 1a (%) | ee of 1a (%) |
|---|---|---|---|---|---|---|
| 1 | 3.5 | 1 | 30 | 7 | 100 | 4 |
| 2 | 3.5 | 1 | 40 | 7 | 99 | 15 |
| 3 | 3.5 | 1 | 40 | 8 | 100 | 8 |
| 4 | 7.0 | 2 | 40 | 7 | 97 | 83 |
| 5 | 35 | 5 | 40 | 7 | 96 | 94 |

Instead, it simply returns the starting material **3**, ready to undergo another redox cycle. Crucially, however, 50% of those triol molecules **3** that return from a fully enantioselective oxidation–reduction cycle have undergone net 180° anticlockwise rotation of the upper ring during the racemization of the transient monoaldehyde intermediate **4**. Thus, after one oxidation–reduction cycle, a proportion of triol **3** is returned in the form of a chemically indistinguishable (degenerate) rotamer in which the hydroxymethyl groups have exchanged positions by means of a 180° anticlockwise rotation. Oxidation of the second hydroxymethyl group (shown in purple) of **3** allows this rotated fraction of the starting material to enter the redox cycle a second time, and again a fraction will undergo 180° anticlockwise rotation. The dissipative cyclic redox reaction network thus no longer supports a deracemization; instead, it drives continuous autonomous net directional motion. We shall now describe how this design for a redox-driven molecular motor was reduced to practice.

The biphenyl rotary motor requires a symmetrically substituted phenyl 'rotor' ring with hydroxymethyl groups in each *ortho* position and a phenolic 'stator' ring substituted by R$^1$ in the final *ortho* position. The nature of substituent R$^1$ allows the rotational barriers of the reduced and oxidized states of the motor (triol **3** and monoaldehyde **4**) to be adjusted for optimal function. We identified *ortho*-fluoro triol **3a** (R$^1$ = F), readily prepared through a five-step synthetic sequence, as a promising motor candidate with a configurationally stable reduced state **3a** and a configurationally labile oxidized state **4a** (Fig. 1d). Rotational barriers of **3a** and **4a** were estimated using desymmetrized derivatives[34]. An enantioenriched sample of silyl ether **5a** showed no detectable racemization after two days at 100 °C in toluene ($\Delta G^{\ddagger}_{rot} > 136$ kJ mol$^{-1}$; Supplementary Information Section 7.3). Acylation of **4a** with Mosher's acyl chloride **7** gave ester **6a** (ref. 35), whose axial diastereoisomers show resolved signals by $^1$H and $^{19}$F nuclear magnetic resonance (NMR) spectroscopy. $^{19}$F NMR exchange spectroscopy analysis of the mixture of diastereoisomers of **6a** indicates that the resolved signals undergo chemical exchange and reveals a barrier to rotation $\Delta G^{\ddagger}_{rot} \approx 79$ kJ mol$^{-1}$ at 40 °C in 2:1 DMSO:D$_2$O (Supplementary Information Section 7.3).

A cyclic reaction network that would allow concurrent enantioselective oxidation of achiral **3a** and non-selective reduction of **4a** was developed by using the cyclic deracemization of chiral analogue **1a** as a model. Alcohol dehydrogenases (ADHs)[36] have been used by Kroutil and co-workers in cyclic reaction networks leading to the deracemization of point-chiral alcohols[37,38] and have also been used in the enantioselective synthesis of chiral biaryls through both desymmetrization and dynamic kinetic resolution processes[39–43]. Inspired by these studies, as well as the seminal report of concurrent biocatalytic oxidation and chemical reduction pathways in the deracemization of benzylic amines by Turner and co-workers[23], we screened a library of ADHs using NADP as cofactor, an NADPH oxidase (YcnD) as the cofactor recycling enzyme[44] and molecular oxygen as the terminal oxidant. Adding ten equivalents of ammonia borane at the outset of the ADH-catalysed oxidation revealed ADH 291 as an effective catalyst for the deracemization of **1a** (Fig. 2). Optimization of the cofactor recycling system, pH and temperature provided ($R_a$)-**1a** in 96% yield and 94% enantiomeric excess (ee) after 24 h (Fig. 2a, entry 5; see Supplementary Information Section 5 for full optimization details). Monitoring the reaction mixture over time (Fig. 2b) established that the ee of **1a** increased over about 24 h as **1a** underwent repeated cycles of oxidation and reduction.

Critically, the deracemization of **1a** confirms the operation of the proposed redox cyclic chemical reaction network: enantiomeric enrichment can arise only through the continuous concurrent operation of the oxidation and the reduction pathways between **1a** and **2a**, which are not the microscopic reverse of each other. Also, this highly effective deracemization of **1a** provides insight into the hierarchy of rates that characterizes the cyclic reaction network: for deracemization to occur, it is necessary that the interconversion of the enantiomeric conformers of **2a** (that is, the enantiomerization of **2a**) occurs faster than the reduction of **2a** back to **1a**. Both the reduction and enantiomerization of **2a** must also occur faster than the oxidation of **1a**. These kinetic constraints can be expressed as a hierarchy of rates, $r_{enant} > r_{red} > r_{ox(Ra,Sa)}$. Furthermore, the deracemization of **1a** confirms that the oxidation is stereoselective, that is, $r_{ox(Sa)}/r_{ox(Ra)} \neq 1$.

Having established an effective deracemization of **1a**, we set about constructing an analogous cyclic redox reaction network under which motor candidate **3a** would undergo rotary motion. We found ADH 291 to be an effective catalyst for the oxidation of triol **3a** to monoaldehyde **4a**. **3a** was treated with ADH 291 under the conditions of the cyclic reaction network: addition of ten equivalents of ammonia borane at the outset of the reaction resulted in a reaction mixture comprising exclusively **3a** after 48 h (Fig. 3a). When ammonia trideuteroborane (H$_3$N·BD$_3$) was used in place of ammonia borane, deuterium was incorporated at the benzylic positions of **3a** (Fig. 3b; see Supplementary Information Section 8.3 for details), confirming that, under these conditions, chemically unchanged motor **3a** is undergoing several cycles of oxidation and reduction.

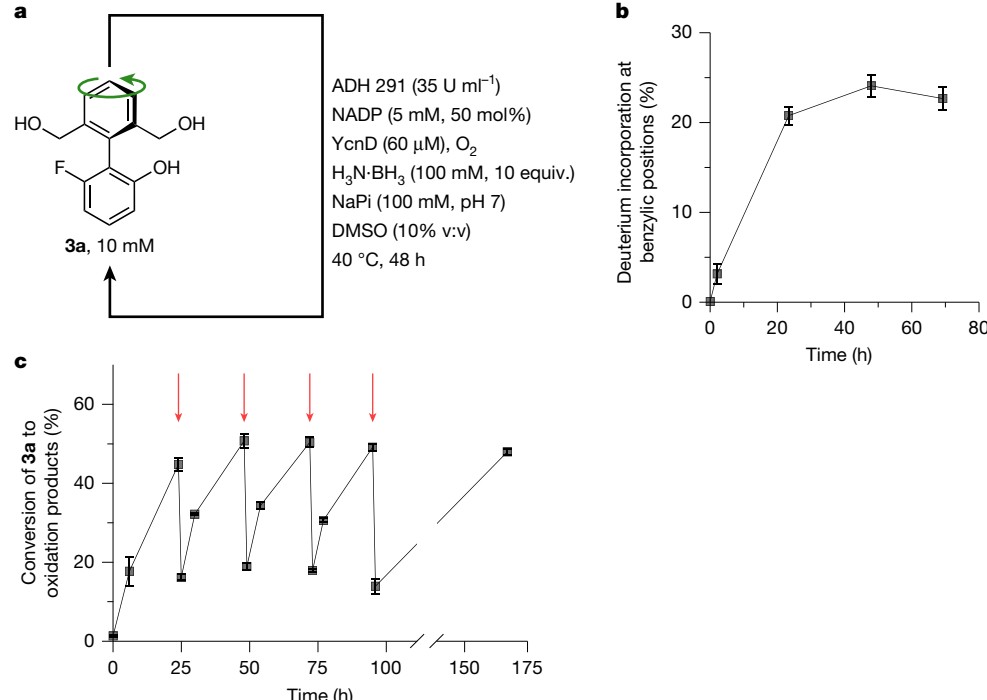

**Fig. 3 | Operation of motor 3a under the cyclic redox reaction network.**
**a**, Motor **3a** under the conditions of the cyclic redox reaction network: **3a** (10 mM), ADH 291 (35 U ml⁻¹), NADP (5 mM, 50 mol%), YcnD (60 µM), $H_3N \cdot BH_3$ (100 mM, 10 equiv.), $O_2$ (air), NaPi (100 mM, pH 7), DMSO (10% v:v), 40 °C, 48 h.
**b**, Ammonia trideuteroborane ($H_3N \cdot BD_3$) in place of ammonia borane ($H_3N \cdot BH_3$) in the cyclic redox reaction network leads to deuterium incorporation at the benzylic positions of **3a**; error bars are derived from triplicate reactions, in which the error given is the population standard deviation between the triplicates.

**c**, Conversion of **3a** to all oxidation products (see Supplementary Information Section 8.4) over time during biocatalytic oxidation of **3a** with the addition of intermittent pulses of $H_3N \cdot BH_3$: **3a** (10 mM), ADH 291 (35 U ml⁻¹), NADP (5 mM, 50 mol%), YcnD (60 µM), NaPi (100 mM, pH 7), $O_2$ (air), DMSO (10% v:v), pulses of $H_3N \cdot BH_3$ (2 mM, 0.2 equiv.) added at 24, 48, 72 and 96 h (addition time points indicated with red arrows); error bars are derived from triplicate reactions, in which the error given is the population standard deviation between the triplicates.

The continued viability of the cyclic redox reaction network over extended periods of time was confirmed by subjecting **3a** to sub-stoichiometric pulses of ammonia borane at regular intervals, rather than a large excess at the outset of the reaction (Fig. 3c; see Supplementary Information Section 8.4 for details). At the outset of the reaction, **3a** was subjected to the standard oxidation conditions, and after 24 h, a combined yield of 45% of all oxidation products was observed. A pulse of ammonia borane (approximately 0.2 equivalents) was added at 24 h, and analysis after 1 h (16% yield of all oxidation products) and a further 24 h (51% yield of all oxidation products) confirmed that the oxidation system (ADH 291, NADP, YcnD and $O_2$) was viable for at least 24 h. A further three pulses of ammonia borane (3× approximately 0.2 equivalents) were applied at 24-h intervals. After each pulse, the yield of all oxidation products returned to a similar value (approximately 50%), confirming the viability of the oxidation system over at least 96 h.

The experiments detailed in Fig. 3 confirm that **3a** is a substrate for the cyclic redox reaction network, that **3a** undergoes several sequential cycles of oxidation and reduction under the standard operating conditions (that is, $[H_3N \cdot BD_3]_0 = 100$ mM) and that the oxidation system remains viable over at least 96 h. Directional rotation of the motor also requires the rate of enantiomerization of **4a** to be greater than the rate of its reduction (that is, $r_{enant} > r_{red}$); if this condition is not met, the motor oscillates between the two redox states **3a** and **4a** but without accompanying rotation about the biaryl axis. Furthermore, effective motor operation requires that the rate of oxidation of **3a** is the slowest of the three constituent steps ($r_{enant} > r_{red} > r_{ox(Ra,Sa)}$) and that the oxidation of **3a** proceeds stereoselectively, that is, $r_{ox(Sa)}/r_{ox(Ra)} \neq 1$. For the deracemization of **1a**, the emergence of enantiomeric enrichment proved that these kinetic constraints were met. However, directional rotation of **3a** has no equivalent stereochemical consequence,

so, to confirm that the motor is operating, we evaluated these rates independently.

To confirm the hierarchy of rates, $r_{enant} > r_{red} > r_{ox(Ra,Sa)}$, we determined each rate separately under conditions matching as closely as possible the operating conditions of the motor. The rate of enantiomerization, $r_{enant}$ (that is, the rate of interconversion of $(R_a)$-**4a** and $(S_a)$-**4a**), was estimated from the barrier to rotation about the equivalent biaryl axis of the Mosher's ester derivative $(S)$-**6a** in a mixture of $D_2O$ and DMSO: $r_{enant} = 4.2 \times 10^{-1} \cdot [\textbf{4a}]$ mM s⁻¹ (Fig. 1d). The rate of the reduction of **4a**, $r_{red}$, was determined by monitoring the reduction of **4a** to **3a** in a NaPi (100 mM, pH 7)/DMSO (10% v:v) mixture at 313 K using ultraviolet–visible spectroscopy (see Supplementary Information Section 7.2 for full details) to give $r_{red} = 2.1 \times 10^{-3} \cdot [H_3N \cdot BH_3] \cdot [\textbf{4a}]$ mM s⁻¹. The undetectably low concentration of **4a** at the steady state of the cyclic redox reaction network ($r_{red} \gg r_{ox(Ra,Sa)}$; see below) precludes the determination of absolute values of $r_{enant}$ and $r_{red}$. However, rotary motor operation is contingent only on the relative rates of $r_{enant}$ and $r_{red}$ and under the standard operating conditions (that is, $[H_3N \cdot BH_3]_0 = 100$ mM), $r_{red}/r_{enant} = 0.50$ (<1, as required; see Supplementary Information Section 7 for details). The rate of oxidation of **3a** ($r_{ox(Ra,Sa)}$) was determined by monitoring the oxidation of **3a** to **4a** under the operating conditions of the motor using high-performance liquid chromatography. In the initial stages of the oxidation reaction, [**3a**] and [$O_2$] are constant and [**3a**] $\gg$ [ADH]. The resulting pseudo-zero-order plot gives $r_{ox(Ra,Sa)} = 5.59 \times 10^{-4}$ mM s⁻¹. Without absolute values for $r_{enant}$ or $r_{red}$, we cannot compare them with this value numerically, but we make the assumption that $r_{red} \gg r_{ox(Ra,Sa)}$, because **3a** is the resting state of the cyclic redox reaction network. The pseudo-zero-order reaction kinetics noted in the determination of $r_{ox(Ra,Sa)}$ remain valid during the operation of the cyclic redox reaction network because the concentration

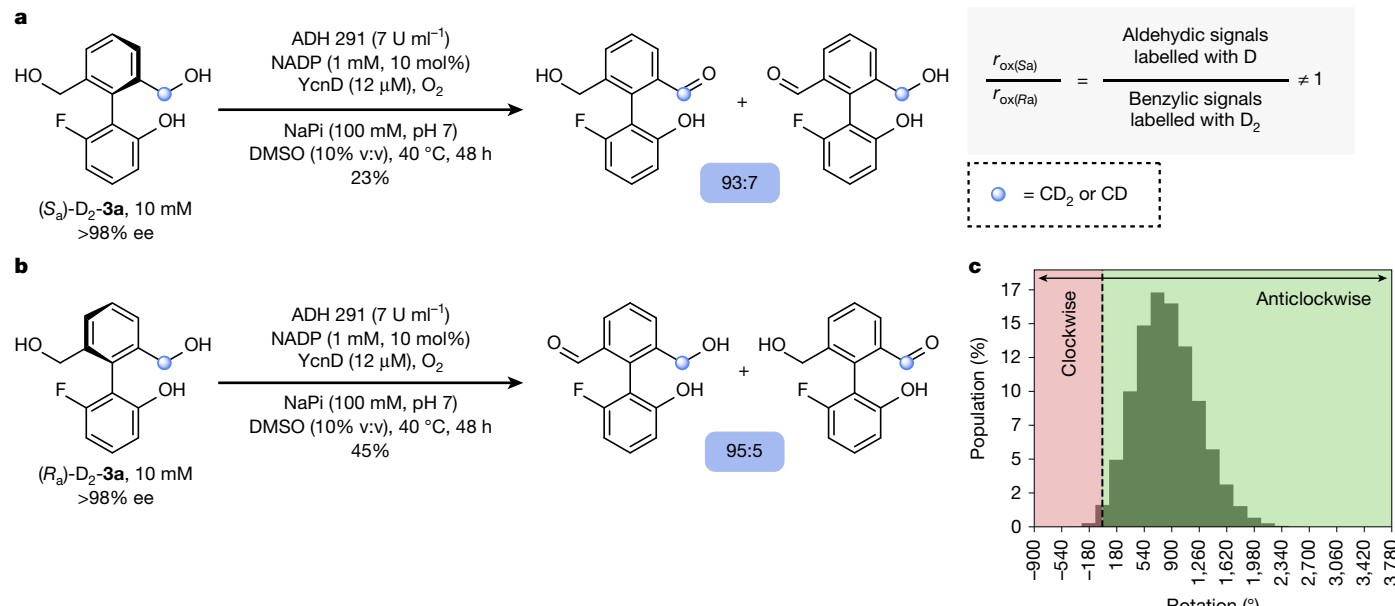

**Fig. 4 | Proof of directional rotation for motor 3a. a**, Oxidation of $(S_a)$-D$_2$-**3a** (($S_a$)-D$_2$-**3a** (10 mM, >98% ee), ADH 291 (7 U ml$^{-1}$), NADP (1 mM, 10 mol%), YcnD (12 μM), O$_2$ (air), NaPi (100 mM, pH 7), DMSO (10% v:v), 40 °C, 48 h), analysis of deuterium distribution at aldehydic CDO and benzylic CD$_2$OH positions and example determination of $r_{ox(Sa)}/r_{ox(Ra)}$. **b**, Oxidation of $(R_a)$-D$_2$-**3a** (($R_a$)-D$_2$-**3a** (10 mM, >98% ee), ADH 291 (7 U ml$^{-1}$), NADP (1 mM, 10 mol%), YcnD (12 μM), O$_2$ (air), NaPi (100 mM, pH 7), DMSO (10% v:v), 40 °C, 48 h) and analysis of deuterium distribution at benzylic CD$_2$OH and aldehydic CDO positions. **c**, Histogram illustrating the distribution of the net angles of rotation of $10^7$ simulated molecules of **3a** after 48 h of operation (see Supplementary Information Section 11 for details).

of **3a** remains constant, being rapidly replenished by the excess of ammonia borane. Under the standard operating conditions of the motor, the rates of the three constituent processes that underpin rotary motor operation do thus indeed conform to the required hierarchy $r_{enant} > r_{red} > r_{ox(Ra,Sa)}$.

The final proof that **3a** undergoes continuous directional rotation under the conditions of the cyclic redox reaction network requires evidence that the oxidation of **3a** proceeds stereoselectively, that is, that $r_{ox(Sa)}/r_{ox(Ra)} \neq 1$. This information cannot be provided by direct observation of enantiomeric enrichment in either starting material or product, because **3a** is achiral and $(R_a)$-**4a** and $(S_a)$-**4a** racemize too fast for analysis of the enantiomeric ratio. Enantioselectivity can nonetheless be deduced from oxidations of enantiopure isotopomers $(S_a)$-D$_2$-**3a** and $(R_a)$-D$_2$-**3a**, which were made with 95% deuterium incorporation and known absolute configuration by methods described in Supplementary Information Section 2.6. As depicted in Fig. 4a, oxidation of $(S_a)$-D$_2$-**3a** generates monoaldehyde **4a**, in which the deuterium label (CD$_2$ or CD) is distributed between the aldehydic CDO and benzyl alcoholic CD$_2$OH positions in a ratio of 93:7 ($^1$H NMR spectroscopy). Conducting the experiment with the isotopomer $(R_a)$-D$_2$-**3a** gives a ratio of deuterium incorporation at the aldehydic and benzylic alcohol positions of 5:95. The enantioselectivity of the oxidation, $r_{ox(Sa)}/r_{ox(Ra)}$, determines the ratio of deuterium incorporation (for example, for $(S_a)$-D$_2$-**3a**, $r_{ox(Sa)}/r_{ox(Ra)}$ = aldehydic signals labelled with D/benzylic alcohol signals labelled with D$_2$), for which the difference between the ratios and the conversions observed for the two isotopomers arises from the kinetic isotope effect in the oxidation. Together, these experiments indicate that oxidation of unlabelled motor **3a** to monoaldehyde **4a** proceeds with an ee that falls in the range 85.7 ± 6.1% ee to 89.5 ± 2.7% ee, confirming that $r_{ox(Sa)}/r_{ox(Ra)} \neq 1$. This experiment provides the first direct evidence of directional motion in an autonomously operating single-bond rotary motor.

The experiments detailed in Fig. 3, the determination of a hierarchy of rates, and this direct observation of enantioselectivity in one of the constituent steps of the cyclic redox reaction network confirm that, under the standard reaction conditions, biphenyl **3a** undergoes continuous net directional rotation about the C–C single bond. With the oxidation of **3a** to **4a** proceeding at a rate of $r_{ox(Ra,Sa)} = 5.59 \times 10^{-4}$ mM s$^{-1}$ and assuming an average enantioselectivity of 87.6% ee, we determine that, for a statistically relevant population of motor **3a**, the mean number of 360° rotations after 48 h of operation will be 2.35 (Fig. 4c; see Supplementary Information Sections 10 and 11 for details).

The motor is driven by the oxidation of the readily available commercial fuel ammonia borane, which—under the current conditions of motor operation—is accompanied by a high background fuel-to-waste reaction (see Supplementary Information Section 8 for details). Future work will aim to explore the fuel efficiency of the motor and to increase its rotational frequency by optimizing the rate and/or enantioselectivity of the oxidation, by varying the reductant and by developing a cyclic redox reaction network comprising stereoselective oxidation and reduction pathways[37,45].

In summary, rotary motor **3a** undergoes autonomous directional rotary motion about a C–C single bond, powered by a biocatalytic cyclic redox reaction network. Our studies confirm **3a** as a substrate for the cyclic redox reaction network; a kinetic analysis establishes that the system conforms to a hierarchy of rates required for rotary rather than oscillatory motion; deuterium isotopomer studies confirm the stereoselectivity of the biocatalytic oxidation of **3a** and allow, for the first time, the directionality of rotary motion of an autonomously operating single-bond molecular motor to be confirmed. Through this report of the first redox-driven single-bond rotary motor, biocatalysis emerges as a powerful tool for the design and development of autonomously operating chemically fuelled molecular motors.

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

## Data availability

The data that support the findings of this study are available in the paper and its Supplementary Information.

## Code availability

The code that supports the findings of this study is available in the Supplementary Information. The code is also available at GitHub (https://github.com/NODbristol/Nature-manuscript-2024-02-03159B).

**Acknowledgements** We thank the Leverhulme Trust (Research Project Grant RPG-2020-031), the Royal Society (RS; University Research Fellowship to B.S.L.C.; URF/R1/180592 and URF/R/231013), the European Research Council (ERC; Advanced Grant 883786 to J.C.), the Engineering and Physical Sciences Research Council (EPSRC; studentships to O.F.B.W., N.T.O.D. and M.D.O. through the Bristol Centre for Doctoral Training in Technology Enhanced Chemical Synthesis; EP/S024107/1) and the Biotechnology and Biological Sciences Research Council (BBSRC; advanced NMR techniques through awards BB/V019163/1 and BB/W008823/1) for funding, Johnson Matthey for the generous gift of alcohol dehydrogenases and cofactors, P. Lawrence, C. Williams and J.-P. Heeb for support with nuclear magnetic resonance experiments, B. Curchod and J. Mortimer for assistance with time-dependent density functional theory calculations and S. Staniland of AstraZeneca for valuable discussions.

**Author contributions** B.S.L.C. and J.C. conceived the project. J.B., O.F.B.W., T.H.N.B., N.T.O.D. and M.D.O. designed and carried out the experiments. A.J.W. carried out the expression and purification of the NADPH oxidase YcnD. B.S.L.C. and J.C. directed the research. All authors contributed to the analysis of the results and the writing of the manuscript.

**Competing interests** The authors declare no competing interests.

**Additional information**
**Correspondence and requests for materials** should be addressed to Jonathan Clayden or Beatrice S. L. Collins.
