## [Peer Review File · Nature]

Redox-powered autonomous directional C–C bond rotation under enzyme control

Corresponding Author: Dr Beatrice Collins

Editorial Notes: Parts of this peer review have been redacted as indicated

Version 0:

Reviewer comments:

Referee #1

(Remarks to the Author)

The manuscript by Berreur et al. titled, "Redox-powered autonomous unidirectional rotation about a C–C bond under enzymatic control," describes the development of a chemoenzymatic cyclic deracemization of a biphenyl substrate to accomplish unidirectional motion of a simple molecular motor. Here, the concept of unidirectional rotation about a C–C bond is quite interesting! However, my overall impression of this work is that the chemistry is underdeveloped with rigor lacking in the approach. My comments are from the perspective of biocatalysis, and I encourage the authors to consider the following points as they further develop this manuscript:

1. ADHs used in this study were obtained as a part of a commercial kit from Johnson Matthey with well-established chemistry. This set of enzymes is capable of the oxidation of alcohol substrates to the corresponding product in the carbonyl oxidation state. The protein sequences are not provided to the reader, which leads to several challenges, including: (a) the reader cannot think about the structure of the catalyst- this would be analogous to publishing a catalytic method without describing the catalyst, (b) it is not possible to obtain catalyst without buying it from this specific vendor, (c) the concentration and activity of the protein used in these studies is not benchmarked as the protein is a part of a lysate that will vary from batch to batch.
2. The biocatalytic reaction conditions given should specify in what form the protein is used. For example, is 2.5 mg/mL referring to the amount of ADH or the amount of the total lysate that is added to the reaction?
3. The authors are cautioned to not make the type of assumption stated on page S27. "By analogy with their stereochemical preference on model compound S8e, we make the assumption that ADH ■, ADH 20 and ADH ■ favour the oxidation of the enantiotopic pro-Sa benzylic alcohol group of other [1,1'-biphenyl]-2-ylmethanol derivatives, including 3a and 1a"
4. An important aspect of any deracemization study is the careful accounting of the mass balance of the material undergoing deracemization. Here, instead of absolute quantification of the material undergoing the proposed deracemization, a relative comparison to other products formed is done by HPLC. It would be more satisfying to quantify the amount of the material by calibration curve and inclusion of an internal standard.
5. In the kinetic analysis, there are some assumptions made which are hard to stand by without additional data. For example, on page S35, "In the initial stages of the reaction, we consider that [3a] and [O₂] are constant and that [3a] >> [ADH]. The latter implies that ADH ■ is operating at V_{max} according to Michaelis-Menten kinetics. The rate law thus becomes dependent on constant concentrations only and can be reduced to a pseudo-zero order process." If [3a] >> [ADH] is true, it does not mean that the ADH is operating at V_{max}. Multiple substrate concentrations must be evaluated to determine V_{max}. In an enzymatic reaction, higher substrate concentration does not equate to a faster or maximum rate of product formation. Substrate inhibition is commonly observed in enzymatic reactions.

Referee #2

(Remarks to the Author)

This is an excellent paper that describes the continuous, chemically fuelled, directionally biased rotation of the rings of a biaryl compound using chemical transformations mediated by enzymes. Leigh demonstrated continuous directionally biased rotation of a biaryl motor in 2022 and the Clayden-Collins system works in a similar way: kinetic resolution as part of a chemomechanical cycle that converts fuel-to-waste. There are several items of novelty here:

- (i) The mechanism involves a redox process: oxidation of an alcohol to aldehyde and reduction back to the alcohol.
- (ii) The authors use enzymes, rather than conventional chemical reagents, to achieve the kinetic resolution in the chemomechanical cycle.
- (iii) The chemomechanical cycle is a directionally biased 180° rotation which is repeated twice (consuming two lots of fuel) to achieve one directionally biased 360° rotation. Repeating a 180° cycle is reminiscent of 2nd generation Feringa light-driven motors, whereas Leigh's 2022 motor employs a 360° cycle.

The work is well done and I'm confident that, with a large initial input of fuel, the motor rotates a couple of times with directional bias until the fuel runs out. However, the motor is a poor catalyst for the fuel-to-waste reaction ($\text{BH}_3\cdot\text{NH}_3 + \text{O}_2 \rightarrow \text{B}(\text{OH})_3$). The catalysis by the motor is not discussed in the main text but Supplementary Table S8 shows that the amount of $\text{B}(\text{OH})_3$ waste formed in the reaction only increases from 51:100 (entry 11) to 88:100 (entry 1) under the motor-operating conditions. This means that most of the fuel is oxidised under the reaction conditions without involving the motor. When fuelling the motor at the start with 10 equivalents of fuel, ~4-out-of-5 of the fuel molecules react by other pathways. In addition to kinetic asymmetry, the key to making a good molecular motor is that the motor should be a good catalyst for the fuel-to-waste reaction. With Leigh's 2022 carbodiimide-fuelled motor (which accelerates the fuel-to-waste reaction >30-fold) >97% of the fuel reacts via the motor-catalysed pathway. Biomolecular motors are uniformly good catalysts.

A few minor points:

1. In the abstract the authors claim the biaryl substrate is "...the most structurally simple synthetic molecular motor yet reported...". Butyl methyl sulfide has been claimed to be the simplest synthetic molecular motor (<https://www.nature.com/articles/nnano.2011.142>). And it seems to me that the Clayden-Collins motor (3a) has 18 non-hydrogen atoms and 4 functional groups, whereas Leigh's 2022 compound has 17 non-hydrogen atoms and 2 functional groups so at least the authors should clarify on what basis 3a is the most structurally simple.
2. It is hard to follow some of the figures in the paper because the compound structures are not always properly defined. For example, R1 and R2 are not defined for any of the structures in the Fig 1 graphic or figure legend. It is only in Fig. 2 that I can see from the structure of (Ra)-1a that 1a must refer to a structure that has a methyl group for R1 and an F for R2.
3. The fuel is actually $\text{BH}_3\cdot\text{NH}_3$ and O_2 , not just $\text{BH}_3\cdot\text{NH}_3$.
4. The authors correctly point out that for the motor to function the order of rates should be $r_{\text{red}} > r_{\text{ox}}$, which they are under model reaction conditions. However, r_{red} is determined to be 2.4×10^{-3} and r_{ox} is 7.6×10^{-4} , a factor of only 3 between them. Really r_{red} should be much faster than r_{ox} (preferably at least 50-fold faster if the motor is going to do more than 1 or 2 rotations (each rotation requiring two redox events)) otherwise there will be significant slippage or free rotation of the motor through the dialdehyde. Free rotation in the dialdehyde would cause any task performed previously by directional rotation to be undone. In the SI the authors claim that they only detect small amounts of the dialdehyde, but that doesn't seem to be consistent with the determined reaction rates.

Overall, this is an excellent paper and a well-conducted study that addresses an important contemporary challenge in synthetic molecular motors in a novel way.

Referee #3

(Remarks to the Author)

A. This paper for the first time shows the use of an (desymmetrization) oxidation reaction coupled with a non-stereoselective reduction to demonstrate overall rotation of a bi-aryl system. It relies on a careful interplay of reaction rate to allow the process to work successfully

B. The design, although relying on known chemistry, is novel, unique (being the first example) and therefore original.

C. The reaction design is elegant, albeit lacking (see below). The experimental data, including determination of rate constants is carried out at the highest level and the characterization data fully supports the hypothesis.

D. Appropriate

E/F. The weakness of the paper is that a large proportion of the molecule do not undergo rotation at all during a given cycle. This is recognized by the authors that write in the introduction that "from monoaldehyde 4, the concurrent non-selective reduction does not return the enantiomer of the starting triol, because this achiral triol 3 is, by definition, superimposable upon its mirror image. Instead, it simply returns the starting material 3, ready to undergo another redox cycle. Crucially, however, a proportion of those triol molecules 3 that return from the oxidation-reduction cycle have undergone net 180° anticlockwise rotation of the upper ring during the racemisation of the transient monoaldehyde intermediate 4 (with the remainder having undergone no net rotation)." The order of rate constant accurately shows that "enantiomerisation" is faster than reduction. This order of rate is crucial in order for net rotation to take place at all. And the authors show that this requirement is fulfilled. However, it also means that the rotation is inefficient. This is due to a combination of two factors A. with rapid enantiomerisation taking place there will be 50% of each enantiomer of 4 in equilibrium. B. The reduction is non-stereoselective meaning that both enantiomers are reduced equally fast. The authors themselves allude to this issue in the end of the paper. The authors clearly demonstrate the principle of rotation for the first time. The question remains whether

this work, subjectively, is appropriate for publication in Nature. I believe that for the paper to be a true breakthrough the authors must make use of an enantioselective reduction process as well. This would make the rotation more efficient, more "realistic" if you will. Furthermore, using both a stereoselective oxidation and reduction process would allow the absolute reaction rates of both to be much higher than what is achieved at the current level of development. If the authors prefer to publish as is I recommend publication in a less prestigious journal.

G. Appropriate

H. The writing is clear and, for such a complex subject, relatively easy to follow.

Version 1:

Reviewer comments:

Referee #1

(Remarks to the Author)

The revised manuscript from Collins, Clayden and coworkers is improved in some technical aspects and provides some food for thought with respect to the overall significance of the work to the scientific community. The technical aspects that have been improved include quantification of the transformation mass balance and added experimental detail. However, these improvements are overshadowed by two elements: (1) the lack of disclosure of the nature of the catalysts- sequences must be included for a biocatalysis-based method to have significant impact, be built upon, be used in new ways, and (2) a conceptually aligned paper was recently published in *Angew. Chem.* DOI: 10.1002/anie.202410112 which uses the manipulation of the oxidation state of the benzylic position to achieve enantioenrichment of axial biaryl bond. Notably, this report is a dynamic kinetic resolution not a deracemization, but the similarities in approach are important to consider here.

Referee #2

(Remarks to the Author)

See attached PDF

Referee #3

(Remarks to the Author)

Version 2:

Reviewer comments:

Referee #1

(Remarks to the Author)

This revised submission is in response to two points raised in a previous round of review:

1. The need to include sequence information for all biocatalysts used in this study, and
2. The need to improve on contextualization of this work with recent advances in the field.

With respect to point 1, it is great to see that the sequence has been included for one ADH used in this study. This required experiments with an ADH with a sequence that could be publicly disclosed. The authors specify that data communicated in Figure 2 and Section 5 of the SI have been repeated with this enzyme, can the authors confirm that the other experiments reported in the SI were conducted with ADH291? For example, the experiments in SI Section 9 specify the use of ADH20- this sequence is not disclosed.

However, the manuscript still details the screening of an ADH library which is not defined and for which no sequence information is provided.

There is another enzyme used in this study for which a sequence is not given, PRO-NOX(001). This enzyme is listed in the text of the manuscript, the caption of Figure 2, and also throughout the supporting information.

In sum, what is reported still falls short of transparent sequence disclosure.

With respect to point 2, the contextualization of this work with recent related examples has not been improved. Although the work might build upon work from the authors, it is still best-practice to reference the most recent advances in a given area, like the work of Hao et al. In addition, the authors should include a reference to a reported cyclic deracemization of atropisomers from Roos et al. DOI: 10.26434/chemrxiv-2025-6s0hk which provides a broader demonstration of this concept on multiple substrates and provides insight on the connection between the structure of the catalyst and the extent of enantioenrichment through protein engineering to modify catalyst selectivity.

Additional notes:

There are figures in the SI that are difficult to interpret which would benefit from additional labels. For example, Fig. S42 has panels a-f defined in the caption, but there is no labeling of panels a-f in the figure that I can see.

In some chromatograms, peaks are not resolved which compromises the accuracy of the values reported (see again Fig. S42).

There are several places in the manuscript and SI that it appears that data points represent single measurements, where it is standard to report $n > 1$. (see Figure 3b and Figure S34 as representative examples).

Referee #2

(Remarks to the Author)

There are a small number of points in the authors responses and the current draft of the manuscript that I feel are still lacking.

(1) As I pointed out previously, the issue of catalysis is of importance for chemically fuelled molecular motors. The current (very) poor catalytic efficiency according to the experimental data the authors report for the motor is useful for others to understand (for designing their own systems) and should be pointed out explicitly in the manuscript.

"We have added text to the manuscript to acknowledge more clearly the relatively high background fuel-to-waste reaction (page 12, lines 6–8: "The motor is driven by the oxidation of the readily available commercial fuel ammonia borane, which under the current conditions of motor operation is accompanied by a relatively high background fuel to waste reaction (see Supplementary Information Section 8 for details). Future work will aim to explore the fuel efficiency of the motor, ...") and indicate our intention to explore the fuel-to-waste reactions in the context of improving the motor's fuel efficiency in later optimisation of the motor."

The text in blue should explicitly state how poor a catalyst the data they are currently reporting shows the motor to be. The text "relatively high background fuel to waste reaction" should be replaced by "up to 80% of the fuel being converted to waste through the relatively high background fuel to waste reaction".

(2) The authors have carried out new experiments to try and quantify the maximum amount of dialdehyde (that would lead to appreciable slippage of the motor) present during motor operation. This is extraordinarily conscientious of them. I did not request it, I only asked for their previous data/experiments to be reported correctly. They state in the response to authors "We have now undertaken extensive re-calibration of dialdehyde S3e and can confidently report that the concentration of dialdehyde S3e is below the limit of detection (LOD), which we determine to be 59 nM....". I do not wish to be unduly negative about this, but it appears that they only monitored the formation of the dialdehyde after motor operation (after 48 hours) instead of checking for dialdehyde being present during motor operation. Checking there is no dialdehyde present after motor operation does not tell you about the level of dialdehyde present during motor operation in the presence of substantial amounts of unspent redox reagents.

The authors provide histograms in Figure R2 "This histogram confirms that even in this "worst-case-scenario", the amount of dialdehyde has relatively little impact on the directional performance of the motor." Figure R2 has different scales for the X-axis in parts (a) and (b) which makes it look like the motor distribution with 0.003% dialdehyde is almost as narrow as it is with 0% dialdehyde, when it is not.

I do have concerns that changing the operating conditions between the various experiments reported in the manuscript is making the authors over-confident (and gives an incorrect impression to readers) that the motor is operating the same way, at the same speed and with the directionality under all the various conditions the authors use. For example, *IF* dialdehyde was really only present at less than 0.003% of the reaction mixture during conditions where the motor was operating efficiently, then it would imply that the diol was reacting hundreds or thousands of times faster with the enzyme than the mono-ol/mono-aldehyde. But this commercial enzyme has broad substrate specificity so having such a substantial difference in reaction rates for similar benzylic alcohols under a wide range of reaction (i.e. fuelling) conditions would seem unlikely. So I suspect that if there is actually less than 0.003% dialdehyde present during motor operation (which, as I noted, is not what the authors actually measured) it may be under a particular set of reaction conditions and it may not be the case for the other reaction conditions the authors have used for various experiments in the paper and SI.

I am absolutely not asking the authors to do more experiments (I did not ask them to do the additional ones they have done) but I do think they should at least consider whether they wish to moderate their claims regarding what this experiment shows.

(3) On page 11 the authors write that their deuterium incorporation experiments are 'This experiment provides the first direct evidence of directional motion in an operational single bond rotary motor.' and repeat this in the summary "...deuterium isotopomer studies confirm the stereoselectivity of the biocatalytic oxidation of 3a and allow, for the first time, the directionality of rotary motion of a functioning single bond molecular motor to be confirmed.' The claims that there was no direct evidence of directional motion of an operational biaryl motor in previous papers is not true. Figure S17 of Nature 604, 80–85 (2022) provides direct evidence of directional 360° rotation of the rotor about the stator of motor-molecule 1-(6'-chlorophenyl)pyrrole 2,2'-dicarboxylic acid. Treatment of the racemic motor-molecule with diisopropylcarbodiimide (DIC) forms the anhydride; hydrolysis with the (S)-hydrolysis promotor affords an enantiomeric excess showing directional bias in anhydride opening (the observation of an e.e. showing that the carboxylic acid groups cannot pass each other in the diacid form); heating then racemizes the molecule which, as the acid groups cannot pass each other, must be occurring by the rotor

carboxylic acid group passing over the chlorine substituent of the stator, completing 360° directionally biased rotation.

Similar direct evidence appears in Fig. 7 of J. Am. Chem. Soc. 147, 8785-8795 (2025). In Nature 637, 594–600 (2025) a myriad of direct evidence is provided for motor rotation directionally twisting polymer chains to cause contraction (and powered re-expansion) of a polymer gel. Whilst I realise the manuscript currently under consideration appeared in its original form on a preprint server more than a year ago, the final published Nature paper (the latest version of which was revised in April 2025) should not, at least in my opinion, ignore results published in the same journal in January 2025 (and on a preprint server over a year ago) and in JACS.

4. For the same reason and for the obvious benefit of readers, other recent papers on autonomous chemically fuelled biaryl motors should (in my view anyway) be cited: Nature 637, 594–600 (2025); J. Am. Chem. Soc. 147, 8785-8795 (2025); J. Am. Chem. Soc. 147, 10690–10697 (2025); ChemRxiv. 2025; doi:10.26434/chemrxiv-2025-jjt70. This is too many Leigh group papers so I would suggest removing current ref (4) [a general reference on artificial molecular machines: Chem. Soc. Rev. 46, 2592–2621 (2017)] and ref (35) [a reference on a stepwise operated motor, all papers of which have little relevance to how autonomous chemically fuelled motors actually work; J. Am. Chem. Soc. 146, 4467–4472 (2024)].

5. Finally, I reiterate my previous view that this is an excellent paper on an important new synthetic molecular motor system.

In the context of this work, and the potential wider applicability of our approach to new classes of molecular motor, we believe that the great advantage of being able to screen panels of enzymes for activity, and hence the potential to devise new cyclic reaction networks quickly and simply, outweighs the disadvantage of not knowing the structure of the enzymes concerned.

- (b) It is indeed true that repeating the exact conditions of the cyclic reaction network using motor substrate **3a** entails buying the ADH from Johnson-Matthey. However, as touched on above, this point rather obscures an advantage of the use of biocatalysis to drive small molecule motors, which is that screening panels of enzymes is highly enabling and allows rapid identification of optimum catalysts for use with structurally diverse small molecular motor candidates. We envisage that the simplicity of the mechanism that underlies the motor's operation will allow it to be adopted broadly, and an approach that tolerates considerable structural diversity through the screening of commercially available panels of enzymes with established broad specificity will be hugely advantageous. Similar panels of enzymes are available from a number of vendors and sources.
- (c) In response to this point (which is related to the concern raised in point 2 below) we have now changed enzyme loadings reported in the manuscript for the optimized reaction conditions from mg/mL to activity units (U), which also addresses the issue associated with the use of different batches of enzyme.

2. Referee #1 notes that "*The biocatalytic reaction conditions given should specify in what form the protein is used*" and asks whether "*For example, is 2.5 mg/mL referring to the amount of ADH or the amount of the total lysate that is added to the reaction?*"

We agree that we were not as clear as we should have been in the original manuscript. In response to this comment, we have now changed to using activity units (U) rather than mg/mL when referring to the amount of ADH used in the optimized reaction conditions.

3. Referee #1 notes that we are “cautioned to not make the type of assumption stated on page S27.”, specifically, “By analogy with their stereochemical preference on model compound S8e, we make the assumption that ADH ■, ADH 20 and ADH ■ favour the oxidation of the enantiotopic pro-Sa benzylic alcohol group of other [1,1'-biphenyl]-2-ylmethanol derivatives, including 3a and 1a”.

We agree that stereochemical assignments can be difficult to make with absolute certainty, and that while analogies are valuable, they should be used with care. We have therefore now gained several additional pieces of information that significantly strengthen the assumptions in the original manuscript, and independently confirm the stereochemical assignments we made. In the original paper, we assigned the absolute configurations of **1a** and **3a** by using a stereochemical correlation with the known atropisomer **S8e**. Our new results provide verification of the stereochemistry of **1a** and **3a**, firstly by independent asymmetric synthesis using a method of reliable enantioselectivity, and secondly by modelling of an experimental circular dichroism spectrum using time-dependent density functional theory (TD-DFT). This evidence has now been incorporated into the Supporting Information and is detailed below.

Independent asymmetric synthesis:

Enantioselectivity of the deracemisation of 1a: In order to confirm independently the assignment of configuration to the two enantiomers of **1a**, we subjected fluorolactone **S1a** to reduction under Bringman’s seminal “lactone method”: treatment with BH₃·THF in the presence of (*R*)-(+)-2-methyl-CBS-oxazaborolidine [*Acc. Chem. Res.* **34**, 615–624 (2001); *Org. Synth.* **88**, 70–78 (2011)]. This reaction gave **1a** in 54% ee, the major enantiomer formed by CBS reduction being the same enantiomer as that formed in the deracemization of **1a** with ADHs ■, 20 and ■ (as reported in Figure 2 in the manuscript and in Tables S4 and S5 of the Supplementary Information). While **S1a** has not previously been reported as a substrate for Bringman’s reduction, the established sense of enantioselectivity with which the (*R*)-CBS catalyst provides enantioenriched biaryls (see references above) confirms the assignment of (*R_a*) configuration to enantioenriched **1a** produced by deracemisation using ADH ■.

Enantioselectivity of the oxidation of 3a (and hence directionality of the motor): A similar approach, again using Bringmann's 'lactone method', was used to establish with certainty the sense of enantioselectivity with which ADH-█ oxidises **3a**. Silylated hydroxylactone **S3h** was treated with $\text{BH}_3 \cdot \text{THF}$ in the presence of (*R*)-(+)-2-methyl-CBS-oxazaborolidine to provide **5a** in 77% ee. The major enantiomer is assigned S_a configuration on the basis of the reliable enantioselectivity of the 'lactone method', and this enantiomer (S_a)-**5a** eluted first on the chiral stationary phase Whelk-O1. This sample of (S_a)-**5a** allowed us to deduce the absolute configurations of the two enantiomers of D_2 -**5a** obtained through separation by semi-preparative HPLC, and hence to assign the absolute configuration of the two isotopomeric triols that result from desilylation, (S_a)- D_2 -**3a** and (R_a)- D_2 -**3a**. As detailed in the SI, this secure assignment reinforced our conclusion that ADH-█ preferentially oxidises the pro- S_a benzylic alcohol of **3a**.

Circular Dichroism:

The circular dichroism spectrum of enantioenriched **1a** formed by deracemisation using ADH-█ was measured in methanol, and compared with the CD spectrum of (R_a)-**1a** predicted by time-dependent DFT calculations (full details in Supplementary Information 2.1). The shapes of the experimental and modelled spectra are congruent (Supplementary Information Figure S3), supporting our assignment of the absolute configuration of (R_a)-**1a** and providing further confirmation of the validity of the Bringmann 'lactone method' for assigning absolute configuration.

4. Referee #1 notes that “An important aspect of any deracemization study is the careful accounting of the mass balance of the material undergoing deracemization. Here, instead of absolute quantification of the material undergoing the proposed deracemization, a relative comparison to other products formed is done by HPLC. It would be more satisfying to quantify the amount of the material by calibration curve and inclusion of an internal standard.”

Prompted by this referee's comment, we have now adopted a new method for analysis that allows us to report full quantification for all optimised results in the manuscript and Supplementary Information. The new method is described in detail in the Supplementary Information but, to summarise, we now analyse the reaction mixtures by reverse phase HPLC,

rather than normal phase HPLC, allowing us to quantify components of the complete reaction mixture without extraction. Calibration curves are then used to quantify absolute concentrations of all peaks observed in the HPLC traces. In addition, as suggested by Reviewer #1, we have also used an internal standard to further support quantification of **1a** during its deracemization. With this method, we have quantified in full the optimised conditions for both the deracemisation of **1a** (Entry 8 in the table in Figure 2a and the plot in Figure 2b, along with additional discussion in the main text) and the operation of the motor **3a** under the full cyclic reaction network (Figure 3 and associated discussion in the main text). The results confirm that **1a** is present in 99% yield and **3a** is present in 100% yield at 48 hours under the operation of the corresponding cyclic reaction networks.

5. Referee #1 notes that “*In the kinetic analysis, there are some assumptions made which are hard to stand by without additional data. For example, on page S35, “In the initial stages of the reaction, we consider that [3a] and [O₂] are constant and that [3a] >> [ADH]. The latter implies that ADH ■ is operating at V_{max} according to Michaelis-Menten kinetics. The rate law thus becomes dependent on constant concentrations only and can be reduced to a pseudo-zero order process.” If [3a] >> [ADH] is true, it does not mean that the ADH is operating at V_{max}. Multiple substrate concentrations must be evaluated to determine V_{max}. In an enzymatic reaction, higher substrate concentration does not equate to a faster or maximum rate of product formation. Substrate inhibition is commonly observed in enzymatic reactions.*”

We agree that we do not have the data to support our original suggestion that ADH-■ is operating at V_{max}, and we have removed this statement. However, this does not change the validity of the kinetic analysis of the rate of oxidation of **3a**. To clarify this, we have re-worded the whole section on the kinetic analysis in both the manuscript and Supplementary Information. Our method for the determination of the rate of oxidation of **3a** ($r_{\text{ox}(R_a, S_a)}$) remains valid, and $r_{\text{ox}(R_a, S_a)}$ was determined by monitoring the oxidation of **3a** to **4a** using HPLC, where the concentrations of **3a** and O₂ are indeed constant under the operating conditions (**3a** is continuously regenerated through reduction of **4a** and O₂ is in large excess) and [**3a**] >> [ADH]. Using the new method for analysis discussed in point 4 above and detailed in the Supplementary Information, we determine the value $r_{\text{ox}(R_a, S_a)} = 5.59 \times 10^{-4} \text{ mM}\cdot\text{s}^{-1}$.

Referee #2:

1. Although Referee #2's discussion of the mechanism of the motor did not raise a specific concern requiring a response, we would nonetheless welcome the opportunity to comment on Referee #2's statement that “[the] Clayden-Collins system works in a similar way [to the Leigh 2022 system]: kinetic resolution as part of a chemomechanical cycle that converts fuel-to-waste.”. We would like to point out that our system does not in fact rely on kinetic resolution (unlike Leigh's, which does). Our *achiral* motor operates through sequential *desymmetrisation*—racemisation cycles as opposed to the kinetic resolution—racemisation mechanism that underpins the Leigh system.

2. Referee #2 notes that “*the motor is a poor catalyst for the fuel-to-waste reaction (BH₃.NH₃ + formally O₂ to B(OH)₃). The catalysis by the motor is not discussed in the main text but Supplementary Table S8 shows that the amount of B(OH)₃ waste formed in the reaction only increases from 51:100 (entry 11) to 88:100 (entry 1) under the motor-operating conditions. This means that most of the fuel is oxidised under the reaction conditions without involving the motor. When fuelling the motor at the start with 10 equivalents of fuel, ~4-out-of-5 of the fuel molecules react by other pathways. In addition to kinetic asymmetry, the key to making a good molecular motor is that the motor should be a good catalyst for the fuel-to-waste reaction. With Leigh's 2022 carbodiimide-fuelled motor (which accelerates the fuel-to-waste reaction >30-fold) >97% of the fuel reacts via the motor-catalysed pathway. Biomolecular motors are uniformly good catalysts.*”

Referee #2 raises some important points with regard to the conversion of fuel to waste by the motor system, and in the light of these comments we have now undertaken further experiments using quantitative ¹¹B NMR spectroscopy to analyse the fuel-to-waste reaction in more detail. These new results are described in detail in the Supplementary Information, and our key findings are also summarised below.

To place this discussion in context, however, we would first like to address Referee #2's comment that “*the key to making a good molecular motor is that the motor should be a good catalyst for the fuel-to-waste reaction*”. It is important to clarify that a background fuel-to-waste reaction does not have a direct bearing on the operation of the motor, but it does limit fuel efficiency. Thus, if the fuel is abundant, cheap and/or readily available (as is the case for

both O₂ and H₃N·BH₃ [CAS Number: 13774-81-7], the fuels that drive motor **3a**), a background fuel-to-waste reaction is of significantly less concern than if the fuel is expensive or synthetically demanding in terms of time or material resources. To draw on the comparison made by Referee #2 to Leigh's 2022 rotary motor, while the Leigh paper does not provide a detailed synthetic procedure, yield, or economic analysis for the synthesis of the fuel, (bis((*R*)-1-phenylethyl)carbodiimide), it states that the fuel “*was synthesised in accordance with reference S6, [Organometallics 35, 3474–3487 (2016)]*”; this reference describes the synthesis of bis((*R*)-1-phenylethyl)carbodiimide over two steps in 52% overall yield. In addition, any attempt to draw broad conclusions about what makes “*a good molecular motor*” based on the fuel used to support motor operation should take account of the full life cycle of the fuel, for example, to what extent is the fuel used by the motor translated into net directional motion as opposed to oscillatory motion and what is the impact of the waste generated in the fuel-to-waste reaction on the long term viability of motor operation. The relative infancy of the field of chemically fuelled single bond rotary motors (this is only the second autonomously operating chemically fuelled single bond rotary motor reported to date) means that the criteria we use as a community to assess the efficacy of these motors remains underdeveloped and we must be wary of identifying specific aspects of motor operation as being “*key to making a good molecular motor*” at the expense of assessing efficacy over a broader range of criteria.

That said, a re-evaluation of the fuel-to-waste reaction that accompanies operation of **3a** has uncovered some interesting points, on which we elaborate below. While motor **3a** is driven by two fuels, H₃N·BH₃ and O₂, the focus of the fuel-to-waste studies described below is the oxidation of H₃N·BH₃, ultimately to the waste product B(OH)₃, and we have not studied the reduction of O₂ in any detail.

We re-assessed our methodology for monitoring the *in situ* formation of B(OH)₃ and have adopted a quantitative ¹¹B NMR spectroscopy method based on a study conducted by Frenich and Fernández [Analyst 143, 4707–4714 (2018)]. Using this optimised method, we began by studying the fuel-to-waste reaction for the deracemization of **1a**. As explained in the manuscript, studying the deracemization of **1a** has the advantage that the emergence of enantiomeric excess provides unequivocal evidence for the continuous operation of the redox reaction network, which is more difficult to probe during the operation of motor **3a**. We were quickly able to establish that under identical conditions to those reported in the manuscript for the deracemization of **1a**, i.e., H₃N·BH₃, (100 mM), ADH ■ (1.0 U/mL), YcnD (12 μM),

NADP (1 mM), pH 8, 25 °C, the presence of biaryl **1a** increased the rate of formation of B(OH)₃ over a 48 hour time period approximately 5-fold (Entries 1 and 2 below). Under these experimental conditions, (*R*_a)-**1a** was generated in 91.3 ± 0.84% ee.

Entry	1a / 10 mM	[B(OH) ₃] / mM
1	no	4.23 ± 0.35
2	yes	19.37 ± 0.40

In the analogous experiment for the operation of motor **3a**, under conditions identical to those reported in the manuscript, i.e., H₃N·BH₃, (100 mM), ADH (35 U), YcnD (60 μM), NADP (5 mM), pH 7, 40 °C, the presence of biaryl **3a** also increased the rate of formation of B(OH)₃ over a 48 hour time period but by a smaller factor (Entries 3 and 4 below). We attribute this less significant acceleration in the fuel-to-waste reaction to a faster background rate of oxidation of H₃N·BH₃ under these more forcing conditions (i.e., higher temperature and loading of ADH; compare Entries 1 and 3).

Entry	3a / 10 mM	[B(OH) ₃] / mM
3	no	41.81 ± 0.83
4	yes	51.13 ± 3.48

In this first generation motor system, we have focused on maximising motor operation as measured by stereoselectivity and chemoselectivity. However, the simplicity of the motor's design, the existence of the deracemization as a “probe” for cyclic reaction network operation, our ability to screen panels of enzymes for activity, and the considerable technological tools

available for enzyme optimization all hold considerable promise for the development of future rotary molecular motors with broad efficacy over a wider range of criteria.

3. Referee #2 queries our characterisation of motor **3a** as “...*the most structurally simple synthetic molecular motor yet reported...*”. Referee #2 then goes on to note that this claim is not supported if the definition of simplicity encompasses the number of non-hydrogen atoms, or the number of functional groups present in the motor.

This is a valid point and we have removed this sentence from the manuscript. “Simplicity” is clearly a subjective term, and our use of it, deriving principally from the achiral nature of **3a**, was ambiguous. We have re-worded that sentence to read: “*Here we show that a redox reaction network, comprising concurrent oxidation and reduction pathways, can drive chemically fuelled continuous autonomous unidirectional motion about a C–C bond in a structurally simple synthetic molecular motor based on an achiral biphenyl.*”.

4. Referee #2 notes that “*It is hard to follow some of the figures in the paper because the compound structures are not always properly defined. For example, R1 and R2 are not defined for any of the structures in the Fig 1 graphic or figure legend. It is only in Fig. 2 that I can see from the structure of (Ra)-1a that 1a must refer to a structure that has a methyl group for R1 and an F for R2.*”.

We acknowledge that there was considerable ambiguity in the definition of R¹ and R² in Figure 1 and the associated legend and thank Referee #2 for drawing our attention to this. We have revised Figure 1 in the manuscript in light of Referee #2’s comments. Details of these revisions are provided in the line edits document. We have also introduced additional Newman projections into Figure 1c, which we believe helps to further clarify the design of unidirectional rotary molecular motor **3**.

5. Referee #2 notes that “*The fuel is actually BH₃.NH₃ and O₂, not just BH₃.NH₃.*”.

This is completely correct, and it was an oversight on our part not to include O₂ in the schemes. All schemes have been revised so that O₂ is included in the reaction conditions and clarification

has been added to the text where appropriate. Details of these revisions are provided in the line edits document.

6. Referee #2 states that *“The authors correctly point out that for the motor to function the order of rates should be renant >rred>rox, which they are under model reaction conditions. However, rred is determined to be 2.4×10^{-3} and rox is 7.6×10^{-4} , a factor of only 3 between them. Really rred should be much faster than rox (preferably at least 50-fold faster if the motor is going to do more than 1 or 2 rotations (each rotation requiring two redox events)) otherwise there will be significant slippage or free rotation of the motor through the dialdehyde. Free rotation in the dialdehyde would cause any task performed previously by directional rotation to be undone. In the SI the authors claim that they only detect small amounts of the dialdehyde, but that doesn't seem to be consistent with the determined reaction rates.”*

We believe that this comment arises from a misunderstanding. The figure 2.4×10^{-3} is the rate constant for the reduction of **4a** not the rate. The rate of this second order reduction reaction is instead given by $r_{\text{red}} = 2.4 \times 10^{-3} \cdot [\text{H}_3\text{N}\cdot\text{BH}_3] \cdot [\mathbf{4a}] \text{ mM}\cdot\text{s}^{-1}$, that is, 2.4×10^{-3} multiplied by the concentration of $\text{H}_3\text{N}\cdot\text{BH}_3$ multiplied by the concentration of **4a**. In contrast, the oxidation is a pseudo-zero order process, and its rate is indeed given by the rate constant, i.e., $r_{\text{ox}(\text{Ra,Sa})} = 7.63 \times 10^{-4} \text{ mM}\cdot\text{s}^{-1}$ (note that r_{red} and $r_{\text{ox}(\text{Ra,Sa})}$ have been revised to $r_{\text{red}} = 2.1 \times 10^{-3} \cdot [\text{H}_3\text{N}\cdot\text{BH}_3] \cdot [\mathbf{4a}] \text{ mM}\cdot\text{s}^{-1}$ and $r_{\text{ox}(\text{Ra,Sa})} = 5.59 \times 10^{-4} \text{ mM}\cdot\text{s}^{-1}$, respectively). As detailed in the manuscript, the undetectably low concentration of **4a** at the steady state of the cyclic redox reaction network precludes the determination of an absolute value for r_{red} and thus direct comparison with $r_{\text{ox}(\text{Ra,Sa})}$. However, as Referee #2 notes, we state that we only detect small amounts of dialdehyde; we have clarified our wording in the manuscript and Supplementary Information and can confirm that we see no dialdehyde when motor **3a** is subject to the cyclic reaction network and, as detailed in our response to point 4 from Referee #1, **3a** is present in a yield of $100\% \pm 3\%$ at 48 hours. As alluded to by Referee #2, this can only be the case if $r_{\text{red}} \gg r_{\text{ox}(\text{Ra,Sa})}$, as required by the hierarchy of rates discussed in the manuscript.

Referee #3:

1. Referee #3 provides a clear and detailed description of the mechanism of motor **3a** and confirms that the manuscript clearly demonstrates the principle of rotation.

Referee #3 then raises the question of whether the work is appropriate for publication in Nature, and correctly identifies that this is subjective. In particular, Referee #3 “*believe[s] that for the paper to be a true breakthrough the authors must make use of an enantioselective reduction process as well*”. It is worth noting that Referee #3 acknowledges that “*the authors themselves alludes to this issue in the end of the paper.*”

We acknowledge Referee #3’s opinion on this point but would like to raise a couple of counterarguments. As noted by Referee #3, we have identified enantioselective reduction as an obvious way to improve the efficiency of motor **3a** (from the manuscript: “*The rotational frequency of biphenyl motor 3a could be increased by ... developing a cyclic redox reaction network comprising stereoselective oxidation and reduction pathways*”). However, we believe that this represents an incremental technical advance to the motor as opposed to a fundamentally novel or innovative design or mechanistic feature. It is highly likely that concurrent stereoselective oxidation and reduction pathways could be developed for motor **3a**, as such cyclic reaction networks have already been reported in the literature (using ADHs) for the deracemisation of point chiral secondary alcohols, as we state in the manuscript: “*Cyclic reaction networks comprising enantioselective biocatalytic reactions for both oxidation and reduction have been reported for the deracemization of point chiral alcohols by Kroutil and co-workers [J. Am. Chem. Soc. **130**, 13969–13972 (2008)]*”.

Significantly, the introduction of a perfectly enantioselective reduction pathway, which is in and of itself unlikely, would only increase the efficiency of the motor by a factor of 2, at best. In contrast, optimisation of the rate of the oxidation pathway holds considerably more scope for rapid access to a more efficient, or as Referee #3 terms it, a more “realistic”, motor.

We do not believe that the technical advance of an asymmetric reduction would provide the reported system with an additional level of novelty and innovation beyond that associated with the reported design. In contrast, our opinion is that the additional complexity of incorporating such a step into the unprecedented motor system described in the current manuscript would significantly detract from its simplicity and impact. We show, for the first time, that ADHs can be combined with a simple reductant, $\text{H}_3\text{N}\cdot\text{BH}_3$. As we hope to disclose in due course, this means that structurally varied motors can be rapidly assessed and optimised through screening of ADHs in combination with this readily available chemical reductant. We believe that the

conceptual and practical simplicity of the current system means that it is also highly accessible to researchers in adjacent fields of research, particularly those who are perhaps not experts in biocatalysis or synthetic chemistry, but who have expertise in supramolecular assemblies or soft matter. We very much hope that the simplicity of our reported system will enable its adoption by researchers looking to exploit molecular motors in more complex environments or molecular architectures and believe that the introduction of an enantioselective reduction has the potential to limit that valuable transferability.

We disagree that “*using both a stereoselective oxidation and reduction process would allow the absolute reaction rates of both to be much higher than what is achieved at the current level of development*”. The efficiency of the motor is only dependent on the rate of the oxidation step (and its stereoselectivity), provided that the rates of the three constituent steps conform to the hierarchy of rates: $r_{\text{enant}} > r_{\text{red}} > r_{\text{ox(Ra,Sa)}}$. An inversion of the relative rates of oxidation and reduction would be disastrous, potentially leading to the formation of overoxidation products and associated uncontrolled rotation (as identified by Referee #2 above). It is also crucial that the reduction remains slower than the enantiomerization of **4a** and any deviation from this would lead to inefficient rotation.

For these reasons we believe that while the introduction of an enantioselective reduction would increase the efficiency of the motor (by a factor of two; as acknowledged in the original manuscript), it would introduce a level of complexity which detracts from the simplicity of the current system. We believe that the future of molecular machines must lie with the development of motors which operate under simple conditions, whose mechanisms can be generally applied to a range of structurally diverse (but synthetically accessible) molecular architectures, which will ensure the best possible opportunities for their uptake and development within functional molecular systems.

Detailed responses to the referees' comments

We have identified the specific concerns of Referees #1 and #2 and numbered them 1–12 below.

Referee #1

1. *“the lack of disclosure of the nature of the catalysts- sequences must be included for a biocatalysis-based method to have significant impact, be built upon, be used in new ways”*

After further discussions with the suppliers Johnson Matthey [JM], we can now report the sequence of the principal enzyme used to drive the molecular motor.

ADH 291. The sequence of this enzyme has been added to the Supplementary Information, and JM will make this enzyme available on request, and this information has also been added to the Supplementary Information.

To ensure that all the data in the paper is associated only with this enzyme of known sequence, we have made some further changes:

- We have repeated the optimisation of the deracemisation of **1a** with ADH 291,

- Given that we can now report both a successful deracemisation and successful motor function with a single enzyme of known sequence, we have removed all data for deracemization and motor function using other commercial enzymes, for which we do not have sequences, from the manuscript and from the Supplementary Information.

- This additional work was carried out by Marc Del Olmo, who has now been added as a co-author to the paper, with the agreement of all other authors.

2. *“a conceptually aligned paper was recently published in Angew. Chem. DOI: 10.1002/anie.202410112 which uses the manipulation of the oxidation state of the benzylic position to achieve enantioenrichment of axial biaryl bond. Notably, this report is a dynamic kinetic resolution not a deracemization, but the similarities in approach are important to consider here.”*

We thank Referee #1 for pointing out this paper, which reports the dynamic kinetic resolution of a configurationally unstable aldehyde by an imine reductase. This paper builds on our previous report on the dynamic kinetic resolution of atropisomeric aldehydes using ketoreductases (*Angew. Chem. Int. Ed.* **55**, 10755–10759 (2016), reference 45 in the manuscript), and complements the growing body of work exploring dynamic kinetic resolution in atroposelective synthesis. However, it does not alter the fact that our manuscript reports the first cyclic deracemization of a chiral atropisomer.

Referee #2

As stated in my original report, this is overall an excellent paper that describes the chemically fuelled, directionally biased rotation of the rings of a biaryl compound using chemical transformations mediated by enzymes. However, I feel the manuscript can be improved by not obfuscating fundamental design flaws (regarding slippage and catalysis) intrinsic to this motor.

We are once again delighted that Referee #2 considers the manuscript to be “*excellent*” and thank them for their detailed reading of the manuscript. Our responses to the points below describe the ways in which we have addressed their comments and suggestions, which we believe further improve the manuscript.

3. *“It’s unfortunate that these new experiments don’t have mass balance in terms of the amount of borane ammonia added. If the amount of other boron species present isn’t measured (only*

B(OH)3, not NH3.BH2(OH) or dimers etc), then one can't tell how many hydride atoms have been consumed and therefore get any real understanding of the efficiency of the motor.”

The concentration of H₃N·BH₃ added in the fuel-to-waste studies has now been included in the relevant table (Table S14 in the Supplementary Information) and we believe this data throws further light on the fuel-to-waste reaction. Inaccuracies in the integration of broad ¹¹B NMR peaks, along with other sampling errors, mean that the concentrations of H₃N·BH₃ and B(OH)₃ do not always sum to 100 mM, but nonetheless no significant quantities of other species are evident. Although mechanisms reported in the literature for the reduction of carbonyl groups with H₃N·BH₃ invoke a number of other boron-containing species (see discussion in *Tetrahedron*, **67**, 7121–7127 (2011) or *Org. Biomol. Chem.*, **18**, 7789–7813 (2020)), we do not detect these species in significant concentrations by ¹¹B NMR at room temperature. Regarding the point about the number of hydrides transferred, studies of the mechanism by which carbonyl groups are reduced by H₃N·BH₃ suggest that H₃N·BH₃ is able to deliver all three hydrides for reduction (*Tetrahedron Lett.*, **21**, 697–700 (1980)). Our ¹¹B NMR spectra show no evidence of partial oxidation or intermediate dehydrogenation products from H₃N·BH₃. We therefore currently assume that all three hydride atoms from each borane species are transferred, either through the reduction pathways of motor **4** or in a background reaction, likely the reduction of NADP.

4. *“To analyse this reaction properly the authors really needed to measure the amount of waste over time – not gather a single data point – to establish if the reaction is zero order, 1st order etc with respect to the ammonia borane, and determine the rate constants.”*

We agree with this referee that a detailed kinetic analysis of the conversion of H₃N·BH₃ to boric acid would be highly desirable. However, the mechanism of the background fuel-to-waste reaction is likely to be highly complex. A plausible primary pathway for this background reaction is the reduction of the co-factor NADP by H₃N·BH₃: it cannot be the direct reaction with the other fuel, oxygen (we have shown that H₃N·BH₃ is stable under the aqueous buffer conditions in the absence of NADP and ADH 291). NADP itself, however, is only present in catalytic quantities (5 mM in the full cyclic reaction network) and is regenerated by the recycling system, YcnD, and the terminal oxidant, oxygen. The pivotal role NADP plays in the frequency-determining oxidation of motor **3a**, further complicates the kinetics. It is also not

yet clear whether either of the two enzymes in the system catalyse the reduction of NADP by $\text{H}_3\text{N}\cdot\text{BH}_3$, and if so, what kinetic regime emerges from the competitive ADH-catalysed reduction of NADP by motor **3a**. Complicating the story yet further, each of the three hydrides transferred from $\text{H}_3\text{N}\cdot\text{BH}_3$ in the course of its overall transformation into $\text{B}(\text{OH})_3$ (see point 3 above) must be transferred through different transition states, presumably each with their own chemoselectivities and relative reaction rates with **3a** and NADP, not to mention different values for k_{cat} and K_{M} . As a result, the major background fuel-to-waste reaction may itself be dependent on the presence or absence of motor **3a**.

Detailed analysis of the immense complexity of this system of competing reactions falls beyond the scope of the current study, but we have aimed to report a few preliminary observations of note. As detailed in Table S14 of the Supplementary Information, we observe a small increase in the formation of $\text{B}(\text{OH})_3$ in the presence of motor **3a**, accompanied by a corresponding increase in the consumption of $\text{H}_3\text{N}\cdot\text{BH}_3$ (see point 3 above). As suggested by Referee #2, we have now also used ^{11}B NMR to monitor the formation of $\text{B}(\text{OH})_3$ from $\text{H}_3\text{N}\cdot\text{BH}_3$ over time under the optimised motor conditions, both in the presence and absence of motor **3a**. Although we were able to extract values both for apparently pseudo-first order rate constants and for initial rates, the same challenges of integrating ^{11}B spectra accurately means that the errors in these values are currently too great to allow us to draw meaningful conclusions.

We have added text to the manuscript to acknowledge more clearly the relatively high background fuel-to-waste reaction (page 12, lines 6–8: “*The motor is driven by the oxidation of the readily available commercial fuel ammonia borane, which under the current conditions of motor operation is accompanied by a relatively high background fuel to waste reaction (see Supplementary Information Section 8 for details). Future work will aim to explore the fuel efficiency of the motor, ...*”) and indicate our intention to explore the fuel-to-waste reactions in the context of improving the motor’s fuel efficiency in later optimisation of the motor. We have promising preliminary data for a different structural class of rotary motor that indicates that the structure of the enzyme and the use of alternative reductants (specifically $\text{Me}_2\text{NH}\cdot\text{BH}_3$) have considerable impact on the rate of the background fuel-to-waste reaction, and these results will inform our optimisation of the motor in future studies.

5. *“In reporting these new experiments, the authors obfuscate how poor a catalyst the experiments show their system is. For the new experiments involving 1a, which is not a motor, in an experiment that uses different conditions (different enzyme, pH, temperature, etc) to the experiment with the actual ‘motor’, they write (p6, lines 21-23): ‘The deracemization of 1a is fuelled by oxidation of ammonia borane, and five times more of the boric acid waste product is formed after 48 h when 1a is present (see Supplementary Information Section 8.5 for details)’. This suggests that 80% of the fuel reacts by the pathway with 1a. But, of course, this experiment is irrelevant in terms of the actual motor catalysis; what matters is the experiment on the actual motor, 3a, using the actual conditions of motor operation. And for this they write only (p8, lines 4- 6): ‘The motor 3a also catalyses an increase in the rate of oxidation of ammonia borane to boric acid (see Supplementary Information Section 8.5 for details)’. In the SI we can see that this actually only corresponds to an increase of ~1.25x B(OH)₃ formed with the motor. So, in fact, under ‘motor’ operation (assuming that it’s 1st order – a big assumption) roughly 80% of the fuel reacts through the background not 20%.”*

It was not our attention to disguise the inefficiency of motor **3a** as a catalyst for the fuel-to-waste reaction of H₃N·BH₃ to B(OH)₃, but rather to make the point that the efficiency appears to be highly dependent on the conditions under which the reaction network is operating. We believe this bodes well for the future optimisation of the fuel efficiency of the motor (see point 4 above for further discussion).

Our revised manuscript now describes the deracemization of **1a** and the operation of motor **3a** using the same enzyme, ADH 291, under the same reaction conditions (see discussion in our response to Referee #1, point 1).

 We have retained only the data corresponding to the operation of motor **3a**, and as detailed in response to point 4, we acknowledge the relatively high background fuel-to-waste reaction in the text of the manuscript.

6. *“A major flaw in this motor design is that when dialdehyde is produced, the motor will spin rapidly in a directionless manner. This is called ‘slippage’ or ‘slip cycles’ in chemical reaction network parlance. Slippage will affect the shape of the distribution of rotations and means that*

Fig 4c is very misleading (see image below). While the centrepoint (mean) of the Gaussian does not change due to slippage, its width does. The current Gaussian has a width (at $y \sim 0$) of 6 full rotations. Assuming 0.01% for the dialdehyde (which undergoes slip), this leads to the width actually being >20 full rotations. A more realistic depiction for Fig 4c. That the authors do not detect dialdehyde in a motor experiment is not sufficient to establish that there is no dialdehyde present nor slippage occurring.

The new Table S20 shows that under conditions where one of the motor alcohols is oxidised in 23-45% yield, 0.5-1% dialdehyde may be formed. Even with a generous estimate for the dialdehyde proportion being, say, no more than 0.1% present during motor operation, there will be lots of slippage in the authors system. During operation the motors would on average spend 0.1% of their time in the dialdehyde form. Over 48 hours of operation this is 170 s. The rate constant for going over the aldehyde barrier they measure as $\sim 1\text{s}^{-1}$, so during its operation the motor will on average slip 170 times backwards/forwards over the barriers. This will cause $\sim 100\text{x}$ more directionless rotations than the total number of directional rotations, which they claim are 1.84 full rotations in 48 hours.”

We fully agree with Referee #2 that formation of the dialdehyde (**S3e**) would be deleterious to the directional performance of the motor due to slippage. However, we have now clarified that some of the assumptions made by Referee #2 are not an accurate reflection of the system, and thus we find that although their concerns are very valid and important to consider, they do not in this instance have a significant impact on the operation of motor **3a** under the experimental conditions described.

Referee #2 uses the data in Table S20 (now Table S19) of the Supplementary Information to make estimates for the quantity of dialdehyde present under the operation of the redox cyclic reaction network (“*The new Table S20 shows that under conditions where one of the motor alcohols is oxidised in 23-45% yield, 0.5-1% dialdehyde may be formed*”). The data in Table S20 (now Table S19), however, corresponds to ‘oxidation-only’ conditions and thus cannot be used to estimate the quantity of dialdehyde formed in the full redox cyclic reaction network, where $\text{H}_3\text{N}\cdot\text{BH}_3$ is present in excess. Considering then the full redox reaction network, Referee #2 states that it would be “*a generous estimate*” that there is “*no more than 0.1% [dialdehyde] present during motor operation*”. It is always challenging to prove the absence of something, and Referee #2 was justified in saying that “*That the authors do not detect dialdehyde in a*

motor experiment is not sufficient to establish that there is no dialdehyde present” based on the experiments we had previously provided.

We have now undertaken extensive re-calibration of dialdehyde **S3e** and can confidently report that the concentration of dialdehyde **S3e** is below the limit of detection (LOD), which we determine to be 59 nM. A concentration of **S3e** of 59 nM would correspond to 0.003% of the reaction mixture, considerably lower than the estimate of 0.1% given by Referee #2. Our determination of the LOD is given below, and this additional analysis shows that under the conditions of operation of motor **3a**, dialdehyde **S3e** is in fact present at a maximum concentration nearly two orders of magnitude lower than the estimate made by Referee #2. We have also provided a revised histogram, using Referee #2’s reasoning, that assumes dialdehyde **S3e** as 0.003% of the reaction mixture. This histogram confirms that even in this “worst-case-scenario”, the amount of dialdehyde has relatively little impact on the directional performance of the motor.

We calibrated dialdehyde **S3e** by decreasing its concentration until **S3e** was undetectable by HPLC. To ensure the lowest possible detection limit we increased the HPLC injection volume by a factor of 5 (from 20 μ L to 100 μ L). Using the standard error of the y-intercept (0.82504 mAu*s) and the gradient (46372 mM/mAu*s) of the calibration curve, we determined the limit of detection (LOD) of **S3e** to be 59 nM, which corresponds to 0.003% of the reaction mixture (*Methods for the determination of limit of detection and limit of quantitation of the analytical methods*, DOI 10.4103/2229-5186.79345). The LOD, generally defined as having a signal-to-noise ratio (S/N) > 3, was verified by examination of the S/N between the signal corresponding to **S3e** (14.8 minutes) and the noise between 12.2–12.8 minutes. Injections were analysed at 488 nM **S3e** and 122 nM **S3e**, where the signals corresponding to **S3e** were easily identifiable (S/N 41.0 and 9.9, respectively), and at 31 nM **S3e**, where the signal could not be confidently distinguished from noise. Motor **3a** was subjected to the conditions of operation for 48 hours (in triplicate) and the yield of **3a** after this time was $99.9 \pm 0.3\%$. In Figure R1 below we provide the HPLC trace of one of these runs, zoomed in on the area of interest for the presence of dialdehyde **S3e** (~14.6 minutes). We used spiking experiments to confirm that the small impurity observed at 14.3 minutes is not dialdehyde **S3e**. From this analysis, we conclude that the concentration of dialdehyde **S3e** is below the LOD, corresponding to less than 0.003% of the reaction mixture.

Figure R1: HPLC trace for the operation of **3a** under the cyclic reaction network after 48 hours; trace is presented having been zoomed in where the signal for dialdehyde **S3e** is expected (14.6 minutes).

We cannot rule out the possibility of the presence of dialdehyde **S3e** below this concentration, but the histograms below (Figure R2) illustrate the limited impact of dialdehyde **S3e** at this concentration on the final rotational state of a 10^7 population of motor molecules.

Figure R2: Histograms illustrating the distribution of the net rotational values of 10^7 simulated molecules of **3a** after 48 h of operation; **a** 0% dialdehyde **S3e**; **b** 0.003% dialdehyde **S3e**.

7. “The authors make a mistake in the assignment of probabilities in their numerical simulation of motor rotation, which leads to them to underestimate the number of directional rotations. They claim the probability that the oxidation of a given motor occurs once during its half-life is $\frac{1}{2}$. This is not true. The probability that the oxidation does not happen during its half-life is $\frac{1}{2}$, but the oxidation could occur multiple times. The number of times an event happens during its half-life is given by the Poisson distribution with mean $\ln(2)$. The probability that the

oxidation happens twice is therefore 0.12. The authors should be able to fix this with minimal changes to their code. I estimate that will increase the number of directional rotations in 48 h from 1.8 to about 2.1.”

We thank Referee #2 for this observation, which is completely correct. We have revised Sections 10 and 11 of the Supporting Information in light of this. In addition, we have corrected an additional mistake in the code, which previously used a value for enantioenrichment when it should have been a value for enantioselectivity. This further increases the number of directional rotations in the 48 hour period. Finally, we have also increased the population size to 10^7 . These changes allow us to conclude that the mean number of 360° rotations after 48 h of operation will be 2.94. The manuscript has been updated to reflect this.

Related to this matter, we have edited one sentence in the manuscript to clarify the operation of the motor. This sentence now read “*Crucially, however, 50% of those triol molecules 3 that return from a fully enantioselective oxidation-reduction cycle have undergone net 180° anticlockwise rotation of the upper ring during the racemization of the transient monoaldehyde intermediate 4.*”.

8. *“If the authors consider the symmetry within their chemomechanical cycle (below), I trust they can appreciate that their system does, in fact, rely on kinetic resolution.”*

The two contrasting rates highlighted by Referee #2 correspond to reactions of enantiotopic groups within a single achiral molecule, rather than the differential rates of reactions of two enantiomeric molecules that would constitute a kinetic resolution. However, there is a conceivable formulation of the chemomechanical cycle as a *dynamic* kinetic resolution if the differential net rates of reduction (rate of reduction minus rate of oxidation) of enantiomeric aldehydes (S_a)-**4** and (R_a)-**4** are considered. We had not considered this alternative view of the cycle, and we thank Referee #2 for their thought-provoking comments.

9. *“It is unfortunate that the authors don’t recognise the fundamental importance of catalysis in chemically fuelled motor design. It is not about aspects of operation or efficacy of performance. A motor needs to transduce energy to perform work and the way an autonomous molecular motor transduces energy from a chemical fuel to perform work is through catalysis*

of the fuel-to-waste reaction. If it is a poor catalyst then, even if the conformational changes are highly directional, the amount of fuel required to perform directional rotations increases exponentially with the number of desired turns and the percentage of chemical energy that can be transduced by the motor to perform work decreases exponentially.”

We do indeed recognise the fundamental importance of catalysis in the operation of chemically fuelled molecular motors: our previous response to Referee #2 sought to comment on the importance of assessing the operation of molecular motors over a broad range of criteria, in response to Referee #2’s use of the term “good”, rather than to underplay the fact that the motor must be acting as a catalyst for the fuel-to-waste reaction. We stand by our previous comments about the importance of an analysis of the full life cycle of a fuel (its availability, price, waste products and their impact on motor operation) in any characterisation of motor operation. We note that a recent publication from the Leigh group, whose 2022 contribution to the field introduced the first and only other example of an autonomously operating chemically fuelled single bond rotary motor [*Nature* **604**, 80–85 (2022)], makes a similar point (“*reflecting the sort of trade-offs in different aspects of performance that are common to both macro-scale motors and biomolecular machinery*” [*JACS* **147**, 8785–8795 (2025)]).

Although Referee #2’s point did not raise a specific concern regarding the manuscript, we wish to reassure Referee #2 that we do indeed understand the importance of the catalytic nature of the motor, and to acknowledge that while the fuelling system that drives the operation of motor **3a** is not yet perfect, it has many valuable characteristics that will contribute to the emerging understanding of motor operation within the field.

10. “*The authors use of the phrase "non-microscopically reverse" is rather unclear. They mean that the borane reduction is not the microscopic reverse of the dioxygen oxidation, which is true, but to the casual reader it probably reads like they are trying to violate microscopic reversibility (all of the reactions are, of course, microscopically reversible).*”

We thank Referee #2 for highlighting the possible ambiguity in our wording regarding the oxidation and reduction pathways of the cyclic reaction network. We have re-worded the relevant sentence, which now reads “*Critically, the deracemization of **1a** confirms the operation of the proposed redox cyclic chemical reaction network: enantiomeric enrichment*

can arise only through the continuous concurrent operation of the oxidation and the reduction pathways between 1a and 2a, which are not the microscopic reverse of each other.”, and believe that this addresses the concerns of Referee #2 on this point.

11. *“The statement (p3, lines 8-10): “A related, but as yet unexplored, model for cyclic deracemization passes not through a transient achiral intermediate (the imine in Figure 1a) but instead through a transient state that consists of a pair of rapidly interconverting enantiomers” is not correct. The Bach deracemizations (Ref. 18, 19, 20) work by exciting the molecule to a state in which there are rapidly interconverting enantiomers.”*

We have modified this sentence in the manuscript, which now reads *“A related, but as yet underexplored, model for thermal cyclic deracemization passes not through a transient achiral intermediate (the imine in Figure 1a) but instead through a transient state that consists of a pair of rapidly interconverting enantiomers.”*, to take into account the existence of related photochemical deracemization processes.

12. *“In the main text Figure 3a, pH should be 7, not 8.”*

We thank Referee #2 for spotting this mistake. This has now been corrected in the manuscript.

Detailed responses to the referees' comments

We have identified the specific concerns of Referees #1 and #2 and numbered them 1–14 below.

Referee #1

1. *“it is great to see that the sequence has been included for one ADH used in this study. This required experiments with an ADH with a sequence that could be publicly disclosed. The authors specify that data communicated in Figure 2 and Section 5 of the SI have been repeated with this enzyme, can the authors confirm that the other experiments reported in the SI were conducted with ADH291? For example, the experiments in SI Section 9 specify the use of ADH20- this sequence is not disclosed.”*

We were pleased to be able to address Referee #1's previous concerns regarding the disclosure of the sequence of the principal enzyme used to drive both the deracemization and the molecular motor. Referee #1 is correct that in Section 9 of the Supplementary Information we have used ADH 20, whose sequence we have not disclosed. ADH 20 is used in Section 9.1 to provide samples of material to validate the NMR experiment we developed to determine the enantioselectivity of the enzymatic oxidation of motor **3a**, which *“cannot be provided by direct observation of enantiomeric enrichment in either starting material or product, because **3a** is achiral, and (Ra)-4a and (Sa)-4a racemise too fast for analysis of the enantiomeric ratio.”* (Supplementary Information, S82). The validation outlined in Section 9.1 is important to ensure confidence in the enantioselectivities we determine from the experiment on **3a** described in Section 9.2 (for which we use ADH 291, whose sequence is disclosed in Section 3). However, this validation could be carried out with material made by any enantioselective method. For example, the enzyme GOase M₃₋₅ has also been reported to oxidise bis-benzylic alcohol **S8e** in high enantioselectivity (99% ee, Staniland, S. *et al.* Enzymatic Desymmetrising Redox Reactions for the Asymmetric Synthesis of Biaryl Atropisomers. *Chem. Eur. J.* **20**, 13084–13088 (2014) [reference 39 of revised manuscript]). Indeed, the validation method described in Section 9.1 could actually be carried out using a different substrate: any configurationally stable symmetrical bis-benzylic alcohol that undergoes stereoselective oxidation (enzymatic or otherwise).

For these reasons, and the fact that ADH 20 is a commercial enzyme whose sequence is protected by IP and is not publicly available, we believe that the use of ADH 20 without disclosure of its sequence, as a means to furnish analytical samples, is justified. Referee #1 previously explained that the “*disclosure of the nature of the catalysts- sequences must be included for a biocatalysis-based method to have significant impact, be built upon, be used in new ways*”; this validation does not represent a biocatalysis-based method that will have significant impact or be built upon and used in new ways, unlike the methods that drive the operation of motor, for which we have now disclosed the sequences of the relevant enzymes.

2. “*However, the manuscript still details the screening of an ADH library which is not defined and for which no sequence information is provided.*”

In the manuscript we state that “*we screened a library of ADHs*”, but in response to Referee #1’s previous concerns during the last round of review regarding the disclosure of sequences for libraries of commercial enzymes, we removed all data corresponding to ADHs whose sequences we are unable to disclose. We maintain, as detailed in our initial response, that “*screening panels of enzymes is highly enabling and allows rapid identification of optimum catalysts for use with structurally diverse small molecular motor candidates*” and that it “*is routine ... to refer to these enzymes by commercial source without further structural information*”. Detailing that this is how we identified the principal enzyme ADH 291 (whose sequence we disclose) is crucial for the reader to understand how we arrived at ADH 291 and we believe that the removal of this discussion from the manuscript would be to its detriment.

3. “*There is another enzyme used in this study for which a sequence is not given, PRO-NOX(001). This enzyme is listed in the text of the manuscript, the caption of Figure 2, and also throughout the supporting information.*”

PRO-NOX(001) is a commercial enzyme whose sequence is protected by IP and is not publicly available. We have removed discussion of PRO-NOX(001) from the text of the manuscript and removed the entry that refers to PRO-NOX(001) (Entry 6) from the Table in Figure 2, along with the associated detail in the Supplementary Information; we believe that the removal of this discussion and entry from Figure 2 does not negatively impact the manuscript but addresses

Referee #1's concerns on this point. We have also removed three entries in Table S7 of the Supplementary Information, which refer to screening undertaken with PRO-NOX(001).

4. *“With respect to point 2, the contextualization of this work with recent related examples has not been improved. Although the work might build upon work from the authors, it is still best-practice to reference the most recent advances in a given area, like the work of Hao et al. In addition, the authors should include a reference to a reported cyclic deracemization of atropisomers from Roos et al. DOI: 10.26434/chemrxiv-2025-6s0hk which provides a broader demonstration of this concept on multiple substrates and provides insight on the connection between the structure of the catalyst and the extent of enantioenrichment through protein engineering to modify catalyst selectivity.”*

We have now included the two references suggested by Referee #1, which were published while the current manuscript was under review. These are references 24 and 43 of the revised manuscript.

5. *“There are figures in the SI that are difficult to interpret which would benefit from additional labels. For example, Fig. S42 has panels a-f defined in the caption, but there is no labeling of panels a-f in the figure that I can see.”*

The panels in Figure S42 of the Supplementary Information should indeed have labels corresponding to the information in the legend and we apologize for this omission. We have now added labels a–f to Figure S42. In addition, we have added labels a and b to Figure S20.

6. *“In some chromatograms, peaks are not resolved which compromises the accuracy of the values reported (see again Fig. S42).”*

While we acknowledge that the resolution of the peaks in panels a–f of Figure S42 is not optimal, we believe it is sufficient to visually validate our spectroscopic method for the determination of the enantioselectivity of the ADH-catalysed oxidation, detailed in Section 9.2.

The key take-away from this method validation experiment is that the obtained enantiomeric excess for unlabelled **S8b** (panels e and f) is clearly higher than that for (*S*_a)-D₂-**S8e** (panels a and b) and lower than that for (*R*_a)-D₂-**S8e** (panels c and d). The errors associated with the relatively poor resolution of the peaks may slightly increase the discrepancy between the HPLC and the NMR results, but do not invalidate the conclusions from this particular experiment nor do they affect the data and conclusions from the experiment described on motor **3a** and its isotopomers (Figure 4 of the main manuscript and Section 9.2 of the Supplementary Information).

7. *“There are several places in the manuscript and SI that it appears that data points represent single measurements, where it is standard to report $n > 1$. (see Figure 3b and Figure S34 as representative examples).”*

In response to Referee #1’s comment, we have repeated the deuterium incorporation experiment in triplicate and the revised data is now plotted in Figure 3b of the manuscript and Figure S33 of the Supplementary Information, with error bars indicating the standard deviation of the data. We have not repeated the experiments detailed in Figure S34 of the Supplementary Information, to which we do not directly refer in the manuscript. The main conclusion that is drawn from the data in Figure S34, that both ADH 291 and YcnD are still operational at 43 h, is independently confirmed (for H₃N·BH₃ in place of H₃N·BD₃ and at reduced H₃N·BH₃ concentration) by the pulse experiment (Figure 3c of the manuscript and Figure S35 of the Supplementary Information) for which a full error analysis based on triplicate experiments of every timepoint is provided.

Referee #2

8. *“As I pointed out previously, the issue of catalysis is of importance for chemically fuelled molecular motors. The current (very) poor catalytic efficiency according to the experimental data the authors report for the motor is useful for others to understand (for designing their own systems) and should be pointed out explicitly in the manuscript.”*

“We have added text to the manuscript to acknowledge more clearly the relatively high background fuel-to-waste reaction (page 12, lines 6–8: “The motor is driven by the oxidation of the readily available commercial fuel ammonia borane, which under the current conditions of motor operation is accompanied by a relatively high background fuel to waste reaction (see Supplementary Information Section 8 for details). Future work will aim to explore the fuel efficiency of the motor, ...”) and indicate our intention to explore the fuel-to-waste reactions in the context of improving the motor’s fuel efficiency in later optimisation of the motor.”

The text in blue should explicitly state how poor a catalyst the data they are currently reporting shows the motor to be. The text “relatively high background fuel to waste reaction” should be replaced by “up to 80% of the fuel being converted to waste through the relatively high background fuel to waste reaction”.

We once again reiterate that we do indeed recognise the fundamental importance of catalysis in the operation of chemically fuelled molecular motors. As detailed in our previous response to related points made by Referee #2, the mechanism of the background fuel-to-waste reaction is likely to be highly complex and due to inaccuracies in the integration of the broad ^{11}B NMR peaks used in the analysis of the fuel-to-waste reaction, we are reluctant at this point to introduce quantified values such as the “80%” suggested by Referee #2, where further analysis, to which we allude in the manuscript (“*Future work will aim to explore the fuel efficiency of the motor*”), is required before such conclusions can be drawn. We do, however, acknowledge Referee #2’s point on this matter and have removed the word “relatively” from the text (the sentence now reads: “*which under the current conditions of motor operation is accompanied by a high background fuel to waste reaction (see Supplementary Information Section 8 for details)*”), which we believe addresses their concerns.

9. *“The authors have carried out new experiments to try and quantify the maximum amount of dialdehyde (that would leads to appreciable slippage of the motor) present during motor operation. This is extraordinarily conscientious of them. I did not request it, I only asked for their previous data/experiments to be reported correctly. They state in the response to authors “We have now undertaken extensive re-calibration of dialdehyde S3e and can confidently report that the concentration of dialdehyde S3e is below the limit of detection (LOD), which*

we determine to be 59 nM....”. I do not wish to be unduly negative about this, but it appears that they only monitored the formation of the dialdehyde after motor operation (after 48 hours) instead of checking for dialdehyde being present during motor operation. Checking there is no dialdehyde present after motor operation does not tell you about the level of dialdehyde present during motor operation in the presence of substantial amounts of unspent redox reagents.”

We would like to clarify that the 48 hour timepoint is not “*after motor operation*”. The choice of the 48 hour timepoint is arbitrary and the pulse experiment (Figure 3c of the manuscript and Figure S35 of the Supplementary Information) shows “*that the oxidation system (ADH 291, YcnD and NADP) remains viable for at least 96 h*”, albeit at lower concentration of $\text{H}_3\text{N}\cdot\text{BH}_3$ —almost twice as long as the 48 hour timepoint used to confirm the absence of dialdehyde in the full CRN in our previous response.

10. “*The authors provide histograms in Figure R2 “This histogram confirms that even in this “worst-case-scenario”, the amount of dialdehyde has relatively little impact on the directional performance of the motor.” Figure R2 has different scales for the X-axis in parts (a) and (b) which makes it look like the motor distribution with 0.003% dialdehyde is almost as narrow as it is with 0% dialdehyde, when it is not.”*

It was in no way our intention to display the additional plots we have provided in Figure R2 as more similar than they are. As this comment does not relate to the manuscript or Supplementary Information, we have taken no further action in response to this comment.

11. “*I do have concerns that changing the operating conditions between the various experiments reported in the manuscript is making the authors over-confident (and gives an incorrect impression to readers) that the motor is operating the same way, at the same speed and with the directionality under all the various conditions the authors use. For example, *IF* dialdehyde was really only present at less than 0.003% of the reaction mixture during conditions where the motor was operating efficiently, then it would imply that the diol was reacting hundreds or thousands of times faster with the enzyme than the mono-ol/mono-aldehyde. But this commercial enzyme has broad substrate specificity so having such a substantial difference in reaction rates for similar benzylic alcohols under a wide range of*

reaction (i.e. fuelling) conditions would seem unlikely. So I suspect that if there is actually less than 0.003% dialdehyde present during motor operation (which, as I noted, is not what the authors actually measured) it may be under a particular set of reaction conditions and it may not be the case for the other reaction conditions the authors have used for various experiments in the paper and SI.”

We start by addressing Referee #2’s broader concerns regarding the differences in “*the operating conditions between the various experiments reported in the manuscript*”. Proving the operation of a molecular motor is famously difficult, in large part because the motor’s behaviour is invariant over time when operating correctly at the fuelled steady state. The individual reactions that make up the cyclic reaction network are “invisible” during operation and in order to confirm its operation we perform experiments which allow us to probe each component separately; crucially, however, we probe these reactions under conditions which mimic as closely as possible that of the operating cyclic reaction network (CRN). There is of course a limit to how closely we can mimic the CRN, but the three experiments we provide in Figure 3 (viz. analysis of the reaction profile of the full CRN at 48 hours, the incorporation of deuterium into the motor over time, and the delayed fuelling experiment over 96 hours), provide convincing support for the claims made regarding motor operation laid out in the manuscript.

To address the specific concerns raised by Referee #2, we can confirm that our analysis claiming that the concentration of dialdehyde **S3e** corresponds to less than 0.003% of the reaction mixture after 48 hours of operation was undertaken using the standard conditions of motor operation. Referee #2 makes a reasonable point by stating “*this commercial enzyme has broad substrate specificity so having ... a substantial difference in reaction rates for similar benzylic alcohols under a wide range of reaction (i.e. fuelling) conditions would seem unlikely*”, but nonetheless their assumption that if the aldehyde is “*present at less than 0.003% of the reaction mixture ... then this [implies] that the diol was reacting hundreds or thousands of times faster with the enzyme than the mono-ol/mono-aldehyde*” is not correct. Discounting the unlikely eventuality that the enzyme is operating at V_{max} (this was discussed in the first round of reviewing), the rate of formation of dialdehyde **S3e** will depend on the concentration of monoaldehyde **4a**, which is orders of magnitude lower than that of triol **3a** at the steady state of the cyclic redox reaction network. The rate of formation of monoaldehyde **4a** is therefore much faster than that of dialdehyde **S3e** not principally because triol **3a** reacts faster with the

enzyme than monoaldehyde **4a**, but instead because there is *much more* triol **3a** than monoaldehyde **4a**. (In fact, in this case it turns out from other experiments that ADH 291 does show chemoselectivity for the triol **3a** over the monoaldehyde **4a**, but this is not a requirement for the system to function).

12. “On page 11 the authors write that their deuterium incorporation experiments are ‘This experiment provides the first direct evidence of directional motion in an operational single bond rotary motor.’ and repeat this in the summary “...deuterium isotopomer studies confirm the stereoselectivity of the biocatalytic oxidation of 3a and allow, for the first time, the directionality of rotary motion of a functioning single bond molecular motor to be confirmed.’ The claims that there was no direct evidence of directional motion of an operational biaryl motor in previous papers is not true. Figure S17 of Nature 604, 80–85 (2022) provides direct evidence of directional 360° rotation of the rotor about the stator of motor-molecule 1-(6'-chlorophenyl)pyrrole 2,2'dicarboxylic acid. Treatment of the racemic motor-molecule with diisopropylcarbodiimide (DIC) forms the anhydride; hydrolysis with the (S)-hydrolysis promotor affords an enantiomeric excess showing directional bias in anhydride opening (the observation of an e.e. showing that the carboxylic acid groups cannot pass each other in the diacid form); heating then racemizes the molecule which, as the acid groups cannot pass each other, must be occurring by the rotor carboxylic acid group passing over the chlorine substituent of the stator, completing 360° directionally biased rotation.”

The example Referee #2 provides as an example of “evidence of directional 360° rotation of the rotor about the stator of motor-molecule” is a stepwise system, which is profoundly different from the autonomous operation of motor **3a** (the stepwise nature of the behaviour of motor-molecule 1-(6'-chlorophenyl)pyrrole 2,2'dicarboxylic acid is clear from the description given by Referee #2 and in the title of the figure referenced by Referee #2 (“Stepwise anhydride formation and hydrolysis of **1b**”, Supplementary Information, Nature **604**, 80–85 (2022)). We have added the term “autonomously operating” as a descriptor to the two instances in the text of the manuscript identified by Referee #2 to clarify our meaning; the two sentences now read: “This experiment provides the first direct evidence of directional motion in an autonomously operating single bond rotary motor.” and “allow, for the first time, the directionality of rotary motion of an autonomously operating single bond molecular motor to be confirmed.”

13. “Similar direct evidence appears in Fig. 7 of *J. Am. Chem. Soc.* **147**, 8785-8795 (2025). In *Nature* **637**, 594–600 (2025) a myriad of direct evidence is provided for motor rotation directionally twisting polymer chains to cause contraction (and powered re-expansion) of a polymer gel. Whilst I realise the manuscript currently under consideration appeared in its original form on a preprint server more than a year ago, the final published *Nature* paper (the latest version of which was revised in April 2025) should not, at least in my opinion, ignore results published in the same journal in January 2025 (and on a preprint server over a year ago) and in *JACS*.”

The direct evidence that appears in Figure 7 of *J. Am. Chem. Soc.* **147**, 8785-8795 (2025) is again for a *stepwise* system and our response to point 12 above applies. As Referee #2 states, “*Nature* **637**, 594–600 (2025) [provides] direct evidence ... for motor rotation directionally twisting polymer chains to cause contraction (and powered re-expansion) of a polymer gel” but this is not the same as direct evidence of directional rotation. We believe that this distinction between direct and indirect evidence of rotational directionality at the molecular level is well understood by the scientific community and this distinction is of considerable importance.

We have added the two references [*Nature* **637**, 594–600 (2025) and *J. Am. Chem. Soc.* **147**, 8785–8795 (2025)], which were published while the current manuscript was under review, to the manuscript. These are references 11 and 12 of the revised manuscript.

14. “For the same reason and for the obvious benefit of readers, other recent papers on autonomous chemically fuelled biaryl motors should (in my view anyway) be cited: *Nature* **637**, 594–600 (2025); *J. Am. Chem. Soc.* **147**, 8785-8795 (2025); *J. Am. Chem. Soc.* **147**, 10690–10697 (2025); *ChemRxiv*. 2025; doi:10.26434/chemrxiv-2025-jjt70. This is too many Leigh group papers so I would suggest removing current ref (4) [a general reference on artificial molecular machines: *Chem. Soc. Rev.* **46**, 2592–2621 (2017)] and ref (35) [a reference on a stepwise operated motor, all papers of which have little relevance to how autonomous chemically fuelled motors actually work; *J. Am. Chem. Soc.* **146**, 4467–4472 (2024)].”

In addition to the two references included in response to point 13 above, we have added the two additional references [*J. Am. Chem. Soc.* **147**, 10690–10697 (2025) and *ChemRxiv*. **2025**; doi:10.26434/chemrxiv-2025-jjt7] to the manuscript. These are references 13 and 14 of the revised manuscript.

Additional Points

15. In our previous analysis of the mean number of rotations of motor **3a** (Supplementary Information Sections 10 and 11 for details) we made the assumption that the racemization of monoaldehyde **4a** reaches completion in every cycle; the motor is more accurately described by a model in which racemization and reduction are competitive processes. To account for this, we have introduced an additional factor to the analysis which depends on the rate of racemization of **4a** at 40 °C, the operating temperature of the motor. The introduction of this factor leads to a small change in the mean number of rotations of the motor in a 48 hour time period from 2.94 to 2.35. The manuscript and Supplementary Information, including the code, have been updated to reflect this correction. We also note that the rate of enantiomerization, r_{enant} , that appears in the manuscript, had been reported at 25 °C instead of 40 °C (the operating temperature of the motor); this has been updated in both documents along with the ratio $r_{\text{red}}/r_{\text{enant}}$ that confirms the hierarchy of rates required for directional rotation of the motor.

As stated in my original report, this is overall an excellent paper that describes the chemically fuelled, directionally biased rotation of the rings of a biaryl compound using chemical transformations mediated by enzymes. However, I feel the manuscript can be improved by not obfuscating fundamental design flaws (regarding slippage and catalysis) intrinsic to this motor.

1. The newly reported experimental data.

We re-assessed our methodology for monitoring the *in situ* formation of $B(OH)_3$ and have adopted a quantitative ^{11}B NMR spectroscopy method based on a study conducted by French and Fernández [*Analyst* **143**, 4707–4714 (2018)]. Using this optimised method, we began by studying the fuel-to-waste reaction for the deracemization of **1a**...

In the analogous experiment for the operation of motor **3a**, under conditions identical to those reported in the manuscript, i.e., $H_3N \cdot BH_3$, (100 mM), ADH (35 U), YcnD (60 μM), NADP (5 mM), pH 7, 40 °C, the presence of biaryl **3a** also increased the rate of formation of $B(OH)_3$ over a 48 hour time period but by a smaller factor (Entries 3 and 4 below). We attribute this less significant acceleration in the fuel-to-waste reaction to a faster background rate of oxidation of $H_3N \cdot BH_3$ under these more forcing conditions (i.e., higher temperature and loading of ADH; compare Entries 1 and 3).

(i) It's unfortunate that these new experiments don't have mass balance in terms of the amount of borane ammonia added. If the amount of other boron species present isn't measured (only $B(OH)_3$, not $NH_3 \cdot BH_2(OH)$ or dimers etc), then one can't tell how many hydride atoms have been consumed and therefore get any real understanding of the efficiency of the motor.

(ii) To analyse this reaction properly the authors really needed to measure the amount of waste over time – not gather a single data point - to establish if the reaction is zero order, 1st order etc with respect to the ammonia borane, and determine the rate constants.

(iii) In reporting these new experiments, the authors obfuscate how poor a catalyst the experiments show their system is. For the new experiments involving **1a**, which is not a motor, in an experiment that uses different conditions (different enzyme, pH, temperature, etc) to the experiment with the actual 'motor', they write (p6, lines 21-23):

'The deracemization of 1a is fuelled by oxidation of ammonia borane, and five times more of the boric acid waste product is formed after 48 h when 1a is present (see Supplementary Information Section 8.5 for details)'. This suggests that 80% of the fuel reacts by the pathway with 1a. But, of course, this experiment is irrelevant in terms of the actual motor catalysis; what matters is the experiment on the actual motor, 3a, using the actual conditions of motor operation. And for this they write only (p8, lines 4-6): 'The motor 3a also catalyses an increase in the rate of oxidation of ammonia borane to boric acid (see Supplementary Information Section 8.5 for details)'. In the SI we can see that this actually only corresponds to an increase of $\sim 1.25x$ $B(OH)_3$ formed with the motor. So, in fact, under 'motor' operation (assuming that it's 1st order – a big assumption) roughly 80% of the fuel reacts through the background not 20%.

2 Slippage/slipping cycles

...we state that we only detect small amounts of dialdehyde; we have clarified our wording in the manuscript and Supplementary Information and can confirm that we see no dialdehyde when motor 3a is subject to the cyclic reaction network and, as detailed in our response to point 4 from Referee #1, 3a is present in a yield of $100\% \pm 3\%$ at 48 hours. As alluded to by Referee #2, this can only be the case if $r_{red} \gg r_{ox}(Ra, Sa)$, as required by the hierarchy of rates discussed in the manuscript.

A major flaw in this motor design is that when dialdehyde is produced, the motor will spin rapidly in a directionless manner. This is called 'slippage' or 'slip cycles' in chemical reaction network parlance. Slippage will affect the shape of the distribution of rotations and means that Fig 4c is very misleading (see image below). While the centre-point (mean) of the Gaussian does not change due to slippage, its width does. The current Gaussian has a width (at $y \sim 0$) of 6 full rotations. Assuming 0.01% for the dialdehyde (which undergoes slip), this leads to the width actually being >20 full rotations.

A more realistic depiction for Fig 4c.

That the authors do not detect dialdehyde in a motor experiment is not sufficient to establish that there is no dialdehyde present nor slippage occurring. The new Table S20 shows that under conditions where one of the motor alcohols is oxidised in 23-45% yield, 0.5-1% dialdehyde may be formed. Even with a generous estimate for the dialdehyde proportion being, say, no more than 0.1% present during motor operation, there will be lots of slippage in the authors system. During operation the motors would on average spend 0.1% of their time in the dialdehyde form. Over 48 hours of operation

this is 170 s. The rate constant for going over the aldehyde barrier they measure as $\sim 1\text{s}^{-1}$, so during its operation the motor will on average slip 170 times backwards/forwards over the barriers. This will cause $\sim 100\text{x}$ more directionless rotations than the total number of directional rotations, which they claim are 1.84 full rotations in 48 hours.

The presence of significant slippage also means that any work done by directional rotation would be undone by random directionless rotation. (Even 0.001% dialdehyde would result in as many directionless rotations as directional rotations and allow any work done by the motor to be undone.) This failure to ratchet the directional bias is an intrinsic flaw in the design and means that work cannot be done cumulatively by such a 'motor'.

3 Statistics of motor rotation

The authors make a mistake in the assignment of probabilities in their numerical simulation of motor rotation, which leads to them to underestimate the number of directional rotations. They claim the probability that the oxidation of a given motor occurs once during its half-life is $\frac{1}{2}$. This is not true. The probability that the oxidation does not happen during its half-life is $\frac{1}{2}$, but the oxidation could occur multiple times.

The number of times an event happens during its half-life is given by the Poisson distribution with mean $\ln(2)$. The probability that the oxidation happens twice is therefore 0.12.

The authors should be able to fix this with minimal changes to their code. I estimate that will increase the number of directional rotations in 48 h from 1.8 to about 2.1.

Some other, minor, points that I hope that the authors will find useful.

- 1. Although Referee #2's discussion of the mechanism of the motor did not raise a specific concern requiring a response, we would nonetheless welcome the opportunity to comment on Referee #2's statement that "*[the] Clayden-Collins system works in a similar way [to the Leigh 2022 system]: kinetic resolution as part of a chemomechanical cycle that converts fuel-to-waste.*". We would like to point out that our system does not in fact rely on kinetic resolution (unlike Leigh's, which does). Our achiral motor operates through sequential desymmetrisation—racemisation cycles as opposed to the kinetic resolution—racemisation mechanism that underpins the Leigh system.

If the authors consider the symmetry within their chemomechanical cycle (below), I trust they can appreciate that their system does, in fact, rely on kinetic resolution.

- 2. Referee #2 notes that “the motor is a poor catalyst for the fuel-to-waste reaction (BH₃.NH₃ + formally O₂ to B(OH)₃)... .. When fuelling the motor at the start with 10 equivalents of fuel, ~4-out-of-5 of the fuel molecules react by other pathways. In addition to kinetic asymmetry, the key to making a good molecular motor is that the motor should be a good catalyst for the fuel-to-waste reaction.” ... if the fuel is abundant, cheap and/or readily available (as is the case for both O₂ and H₃N·BH₃ [CAS Number: 13774-81-7], the fuels that drive motor 3a), a background fuel-to-waste reaction is of significantly less concern than if the fuel is expensive or synthetically demanding in terms of time or material resources.....the criteria we use as a community to assess the efficacy of these motors remains underdeveloped and we must be wary of identifying specific aspects of motor operation as being “key to making a good molecular motor” at the expense of assessing efficacy over a broader range of criteria.

It is unfortunate that the authors don't recognise the fundamental importance of catalysis in chemically fuelled motor design. It is not about aspects of operation or efficacy of performance. A motor needs to transduce energy to perform work and the way an autonomous molecular motor transduces energy from a chemical fuel to perform work is through catalysis of the fuel-to-waste reaction. If it is a poor catalyst then, even if the conformational changes are highly directional, the amount of fuel required to perform directional rotations increases exponentially with the number of desired turns and the percentage of chemical energy that can be transduced by the motor to perform work decreases exponentially.

- The authors use of the phrase "non-microscopically reverse" is rather unclear. They mean that the borane reduction is not the microscopic reverse of the dioxygen oxidation, which is true, but to the casual reader it probably reads like they are trying to violate microscopic reversibility (all of the reactions are, of course, microscopically reversible).

- The statement (p3, lines 8-10): "*A related, but as yet unexplored, model for cyclic deracemization passes not through a transient achiral intermediate (the imine in Figure 1a) but instead through a transient state that consists of a pair of rapidly interconverting enantiomers*" is not correct. The Bach deracemizations (Ref. 18, 19, 20) work by exciting the molecule to a state in which there are rapidly interconverting enantiomers.

- In the main text Figure 3a, pH should be 7, not 8.